# Phenotypic manifestation of α-synuclein strains derived from Parkinson's disease and multiple system atrophy in human dopaminergic neurons

Benedict Tanudjojo[1,7], Samiha S. Shaikh[1,7], Alexis Fenyi[2], Luc Bousset[2], Devika Agarwal[3,4], Jade Marsh[1], Christos Zois[4], Sabrina Heman-Ackah[5], Roman Fischer [6], David Sims[3,4], Ronald Melki[2] & George K. Tofaris [1✉]

α-Synuclein is critical in the pathogenesis of Parkinson's disease and related disorders, yet it remains unclear how its aggregation causes degeneration of human dopaminergic neurons. In this study, we induced α-synuclein aggregation in human iPSC-derived dopaminergic neurons using fibrils generated de novo or amplified in the presence of brain homogenates from Parkinson's disease or multiple system atrophy. Increased α-synuclein monomer levels promote seeded aggregation in a dose and time-dependent manner, which is associated with a further increase in α-synuclein gene expression. Progressive neuronal death is observed with brain-amplified fibrils and reversed by reduction of intraneuronal α-synuclein abundance. We identified 56 proteins differentially interacting with aggregates triggered by brain-amplified fibrils, including evasion of Parkinson's disease-associated deglycase DJ-1. Knock-out of DJ-1 in iPSC-derived dopaminergic neurons enhance fibril-induced aggregation and neuronal death. Taken together, our results show that the toxicity of α-synuclein strains depends on aggregate burden, which is determined by monomer levels and conformation which dictates differential interactomes. Our study demonstrates how Parkinson's disease-associated genes influence the phenotypic manifestation of strains in human neurons.

[1] Nuffield Department of Clinical Neurosciences, University of Oxford, Oxford, UK. [2] CEA, Institut François Jacob (MIRCen) and CNRS, Laboratory of Neurodegenerative Diseases, Fontenay-aux-Roses, France. [3] MRC Centre for Computational Biology, University of Oxford, Oxford, UK. [4] MRC Weatherall Institute of Molecular Medicine, University of Oxford, Oxford, UK. [5] Department of Neurosurgery, University of Pennsylvania, Philadelphia, PA, USA. [6] Target Discovery Institute, Nuffield Department of Medicine, University of Oxford, Oxford, UK. [7] These authors contributed equally: Benedict Tanudjojo, Samiha S. Shaikh. ✉email: george.tofaris@ndcn.ox.ac.uk

The formation of filamentous α-synuclein inclusions is the defining neuropathological characteristic of two clinically distinct conditions, Parkinson's disease (PD) and multiple system atrophy (MSA). PD typically presents with a levodopa-responsive movement disorder, whereas MSA is characterised by a combination of extrapyramidal, cerebellar and autonomic features that do not typically respond to therapy, leading to death within 7–9 years. Neuropathologically, PD is characterised by intraneuronal cytoplasmic and neuritic inclusions termed Lewy bodies (LB) and Lewy neurites (LN), respectively, whereas MSA is characterised by glial cytoplasmic inclusions or neuronal intra-nuclear inclusions and neuropil threads[1,2]. Both conditions share the key pathological feature of degeneration of dopaminergic neurons in the substantia nigra (SN). Unlike PD, α-synuclein pathology in MSA brains is not uniformly detected in dopaminergic neurons, raising the possibility that neurons containing MSA-type α-synuclein assemblies die much faster than in PD. Sarkosyl-insoluble fibrils from the human brain have a higher seeding potency in animal models when isolated from glial cytoplasmic inclusions than LB-isolated α-synuclein, which may account for the more aggressive nature of MSA[3,4]. Whether this happens in human neuronal models is currently unknown.

The association of α-synuclein with diverse clinical phenotypes suggests the existence of distinct structural conformers or strains that may at least partly account for the different symptoms or severity. Recombinant α-synuclein can assemble into diverse de novo-generated amyloid structures that can imprint their conformational characteristics onto naive forms of the protein[5] and exhibit different pathological properties when injected into animals[6,7]. However, it remains unclear how the interplay between α-synuclein levels, aggregate burden or conformation determines neurodegeneration in human neurons and which of these are the molecular determinant of pathology.

## Results

### α-Synuclein levels and conformation determine seeded aggregation in human iPSC-derived dopaminergic neurons.

To investigate whether pathophysiologically relevant levels of wild-type or mutant forms of α-synuclein influence its propensity to aggregate within human dopaminergic neurons, we differentiated induced pluripotent stem cell (iPSC) lines from three healthy controls, two patients carrying the SNCA[A53T] mutation which in vitro accelerates fibrillation[8] and three clones with SNCA triplication (SNCA[TRIP]) which is associated with more severe disease[9] (Supplementary Table 1). Quality control analyses of all the iPSC lines were previously described[10,11]. iPSC lines were differentiated into dopaminergic neurons using a modified floor-plate protocol that we previously optimised[12]. Neurons were used for experiments from days in vitro (DIV) 45 onwards, a timepoint at which we previously found iPSC-derived human dopaminergic neurons to be electrophysiologically active[12]. We confirmed by immunofluorescence that in all lines dopaminergic neurons co-expressed β-3 tubulin (TUJ1) with tyrosine hydroxylase (TH) as shown in Supplementary Fig. 1a. The differentiation efficiency, as assessed in all clones by immunofluorescence for TUJ1 and TH was similar across the genotypes used with ~93% of cells expressing TUJ1 and 76% expressing TH (Supplementary Fig. 1b). Cell-type deconvolution analysis of bulk RNASeq with a reference multi-subject human single-cell atlas of the SN[13] using the MuSiC method[14] estimated the cell-type composition of the bulk RNASeq samples across multiple bootstraps and identified 60–70% dopaminergic neurons in accordance with our immunostaining data (Supplementary Fig. 1c).

To assess whether the addition of exogenous de novo-generated fibrils seed the aggregation of endogenous α-synuclein, we measured pathological phosphorylation of serine 129, referred to herein as pSyn, and visualised the localisation of pSyn-positive inclusions. We found that fibrils induced the accumulation of pSyn aggregates in cell bodies and processes (Fig. 1a, b), which co-localised with p62, a marker of inclusion pathology (Fig. 1b). We used cell fractionation and immunoblotting to confirm that detection of pSyn accumulation corresponded to aggregated α-synuclein in the detergent-insoluble $100,000 \times g$ pellet (Fig. 1c). We found that pSyn is detected almost exclusively in the insoluble fraction (97%) compared to the soluble fraction (Supplementary Fig. 2), indicating that it reflects aggregated α-synuclein. We also showed that fibrils generated from recombinant α-synuclein with the serine 129 to alanine (S129A) mutation that cannot be phosphorylated were equally potent in inducing pSyn aggregates in iPSC-derived neurons, demonstrating that the pSyn signal arises from aggregation of intraneuronal α-synuclein and not phosphorylation of exogenous seeds (Supplementary Fig. 3). We then compared the extent of aggregation across neuronal cultures derived from healthy controls or patients with SNCA[A53T] or SNCA[TRIP] using the FRET assay for pSyn. When corrected to their corresponding non-seeded control, aggregation was highest in SNCA[TRIP] lines followed by SNCA[A53T] and healthy control lines (Fig. 1d). At the timepoints tested, there was no aggregation under non-seeded conditions based on pSyn or oligomeric FRET signal across all lines (Supplementary Fig. 4). Therefore, monomeric A53T mutant α-synuclein or increased wild-type protein levels promote seeded aggregation in human neurons with the latter having the strongest effect. We investigated the evolution of these aggregates with time in culture by comparing healthy control to SNCA[TRIP] lines. Formation of pSyn aggregates in iPSC-derived dopaminergic neurons was detectable at 1 week, increasing further by 4 weeks post seeding (Fig. 1e). This time- and dose-dependence was reminiscent of intraneuronal templating of available monomer and correlated with the progressively increased expression of α-synuclein with time in culture (Fig. 1f). SNCA[TRIP] lines exhibited ~2.5-fold increase in endogenous α-synuclein monomer compared to healthy control neurons at all timepoints tested (Fig. 1f). To determine whether aggregation had an effect on α-synuclein expression, we performed qPCR comparing baseline and 2 weeks post seeding with fibrils. We found that α-synuclein mRNA was significantly increased in both control and SNCA[TRIP] neurons upon seeding (Fig. 1g). This finding suggests that sequestration of α-synuclein monomer into aggregates leads to the compensatory upregulation of SNCA mRNA.

To assess whether α-synuclein structure can also influence the extent of aggregation in human neurons, we compared fibrils with another de novo-generated α-synuclein strain, termed ribbons that were previously shown to exhibit different structural characteristics and seeding propensities[5]. We found that in human iPSC-derived dopaminergic neurons, polymorph ribbons yielded significantly higher levels of pSyn than polymorph fibrils as previously shown in cortical iPSC neurons[15], suggesting that the structure of α-synuclein seeds also determines its propensity to recruit endogenous α-synuclein and promote aggregation (Fig. 1h). Thus, both endogenous α-synuclein protein abundance and α-synuclein seed conformation can accelerate the formation of pSyn-positive aggregates in human dopaminergic neurons.

### α-Synuclein strains are amplified in the presence of human brain homogenate.

To investigate the effects of α-synuclein strains from PD and MSA, we generated synthetic descendants of aggregated α-synuclein in human brain homogenate using protein misfolding cyclic amplification (PMCA). Brain tissue from a total of 11 subjects was used for this study, three cases of sporadic

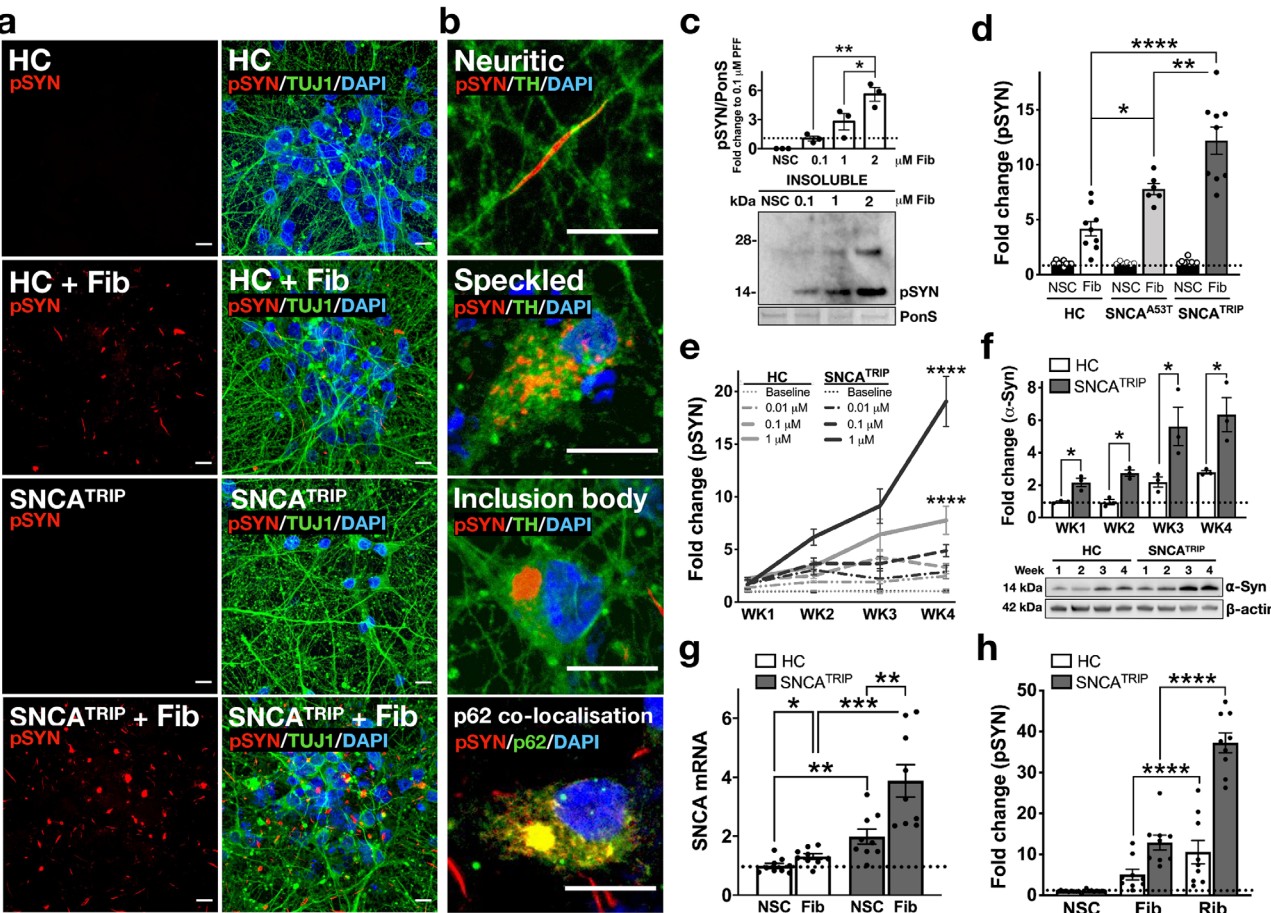

**Fig. 1 Characterisation of α-synuclein seeded aggregation in iPSC-derived dopaminergic neurons. a** Addition of exogenous α-Syn fibrils (+Fib) seeded the aggregation of endogenous α-Syn as evident by pSYN immunoreactivity in red. Images are representative of three independent differentiations (scale bar: 10 µm). **b** pSYN-positive aggregates were localised in axons (neuritic) or soma (speckled or Inclusion body) and co-localised with p62. Images are representative of three independent differentiations (scale bar: 10 µm). **c** RIPA-insoluble fractions from seeded neuronal lysates were immunoreactive for pSYN by immunoblotting (n = 3). **d** Dopaminergic neurons harbouring either the missense A53T mutation (SNCA$^{A53T}$) or SNCA gene triplication (SNCA$^{TRIP}$) had increased levels of pSYN compared to healthy controls 2 weeks post seeding, with SNCA$^{TRIP}$ neurons exhibiting the highest burden of seeded aggregates as quantified by HTRF (n = 9 for HC, n = 9 for SNCA$^{TRIP}$, n = 6 for SNCA$^{A53T}$). **e** A single exposure to fibrils on DIV45 at an increasing dose (baseline, 0.01, 0.1, 1 µM) induced a dose and time-dependent increase in pSYN signal as quantified by HTRF (n = 9 for SNCA$^{TRIP}$ and n = 5 for HC). **f** Endogenous α-Syn expression in neurons increased over time in culture. Increased baseline α-Syn expression by ~2.5-fold in SNCA$^{TRIP}$ neurons compared to controls as assessed by immunoblotting (n = 3). **g** Upregulation of SNCA mRNA was triggered by fibril treatment 2 weeks post seeding at DIV45. Induction of SNCA mRNA was more pronounced in SNCA$^{TRIP}$ compared to controls (n = 9). **h** Ribbons exhibited higher seeding propensity than fibrils 4 weeks post seeding as evident by pSYN levels, quantified by HTRF (n = 9). Each dot corresponds to one clone differentiated once and data are mean ± s.e.m from at least n = 3 differentiations per clone. *P < 0.05, **P < 0.01, ***P < 0.001, ****P < 0.0001 by one-way ANOVA followed by Tukey's multiple comparison test (**c**, **d**, **h**), two-way ANOVA followed by Tukey's multiple comparison test (**e**), two-sided unpaired Student's t test (**f**, **g**). Source data for **c**, **d**, **e**, **f**, **g**, **h** are provided as a Source Data file.

PD, three controls and five MSA cases as summarised in Supplementary Table 2. Anterior cingulate was obtained from each PD and control case, whereas cerebellum was used for each MSA and control case. For one PD case, we also assessed brain tissue from the Temporal Gyrus (labelled PD 3G) as well as the cingulate cortex (labelled PD 3C) in this initial characterisation. The amount of total α-synuclein and pSyn were quantified by filter retention and FRET (Supplementary Fig. 5). These measurements showed that first, the total amount of α-synuclein was slightly lower in MSA patients brain homogenates (Supplementary Fig. 5a) and secondly the amount of pSyn was similar in control and PD patients brain homogenates and significantly higher in homogenates from MSA patient brains (Supplementary Fig. 5b). As expected, the amount of pathogenic, phosphorylated α-synuclein, varied significantly from one patient to another and was 5–12-fold higher in PD and MSA patients brain homogenates

compared to similar homogenates from control patients (Supplementary Fig. 5c). Homogenates were blindly subjected to three cycles of amplification by means of PMCA. Amplification was performed in an iterative manner to avoid de novo assembly of monomeric α-synuclein into fibrils within the experimental timeframe. The assembly of α-synuclein was monitored through thioflavin T binding. At time 400 min, when thioflavin T signal was detectable in the patient brain homogenate but not in the controls (Fig. 2a), an aliquot of the first amplification cycle was diluted (20-fold for MSA patients and control patients, 50-fold for PD patients) in the presence of monomeric α-synuclein (100 µM in assembly buffer) and assembly was further monitored by thioflavin T binding. Two additional amplification rounds were performed by diluting the assemblies generated at the end of amplification cycle 2 and 3 (4 h for PD, MSA and control patients for cycle 2, 3 h for PD patients and 6 h for MSA and control

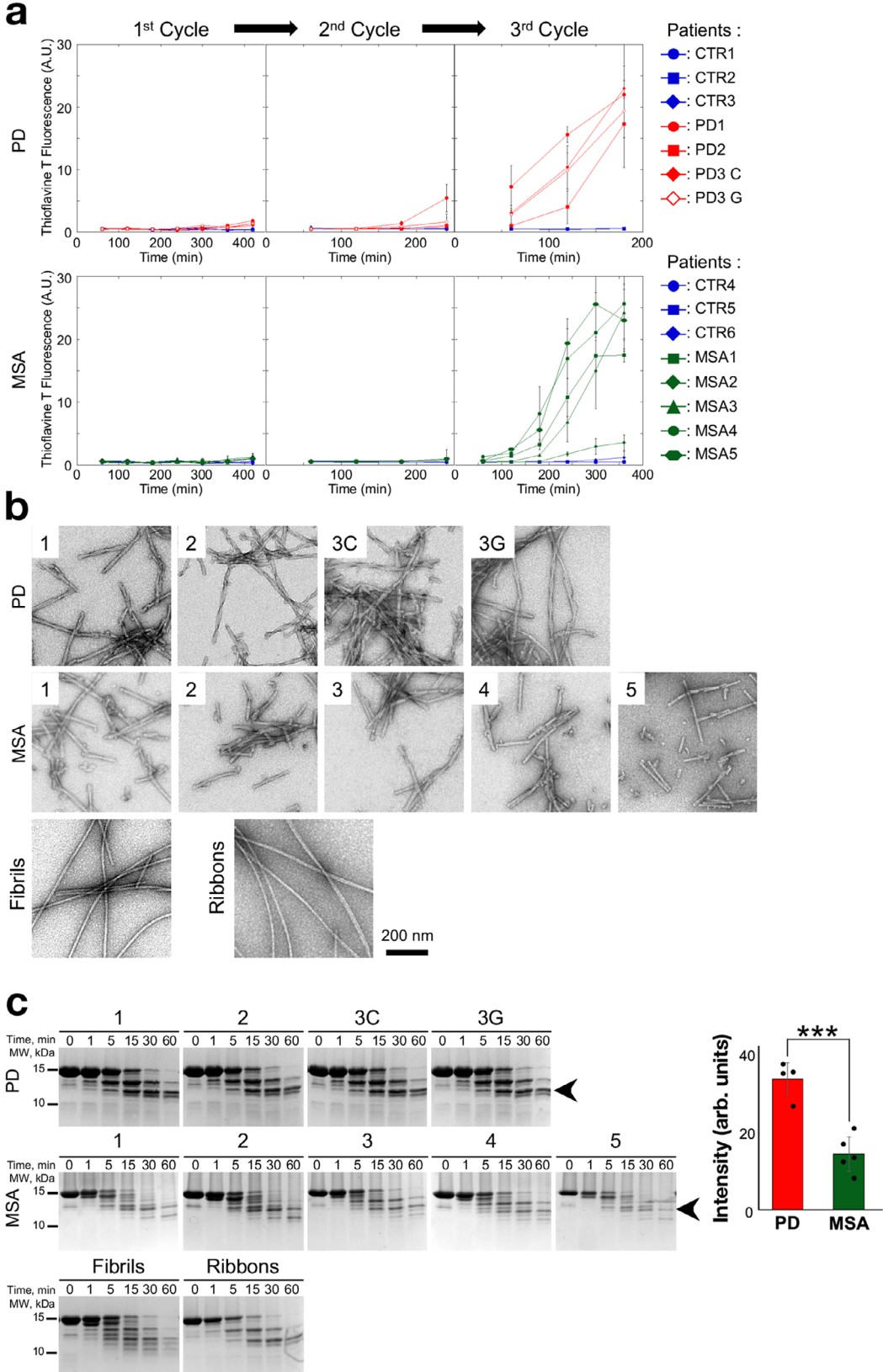

patients for cycle 3) in assembly buffer containing monomeric α-synuclein. We found that homogenates from the three PD and five MSA patients but not the control patients exhibited seeding propensity (Fig. 2a). The resulting α-synuclein assemblies were imaged by transmission electron microscopy (TEM) and finger-printed by SDS-PAGE after limited proteolysis using Proteinase

K. TEM analysis demonstrated that the resulting assemblies are of fibrillar nature (Fig. 2b). Both PD and MSA brain-amplified fibrils exhibited a twisted shape. PD and MSA brain-amplified α-synuclein fibrils exhibited distinct limited proteinase K degradation patterns (Fig. 2c and Supplementary Fig. 6). The proteolytic pattern of the fibrils amplified with PD brain homogenate, unlike

**Fig. 2 Amplification of pathogenic α-synuclein from PD and MSA patients by PMCA and characterisation of the resulting assemblies. a** PMCA was performed on PD (top panels, in red and corresponding controls, blue) and MSA (bottom panels, in green) and the corresponding controls (in blue). Patient brain homogenates (2% w:v) for the first cycle, in PMCA buffer (150 mM KCl, 50 mM Tris-HCl, pH 7.5) containing monomeric α-synuclein (100 µM). The following amplification cycles were seeded by 2 or 5% (v:v) of the preceding amplification cycles for PD and MSA, respectively. The amounts of brain homogenates and PMCA-amplified assemblies used in each amplification reaction were defined through an optimisation study aimed at maintaining high stringency by minimising the de novo aggregation of α-synuclein under the experimental conditions we used. The time at which an aliquot from a given amplification cycle was withdrawn for a subsequent amplification reaction (last timepoint for each amplification cycle) was also defined through an optimisation study aimed at avoiding the formation of de novo α-synuclein fibrillar assemblies. The curves represent an average of $n = 4$ replicates ± SD. **b** Electron micrographs of α-synuclein assemblies obtained after the third cycle of amplification by PMCA from each of the three PD, five MSA cases and de novo-generated α-synuclein fibrils and ribbons. For PD3, we examined the temporal gyrus (PD 3G) as well as the cingulate cortex (PD 3C). Scale bar: 200 nm. **c** Limited proteolytic patterns of α-synuclein assemblies obtained after the third cycle of amplification by PMCA from PD-, MSA- and de novo-generated α-synuclein fibrils and ribbons. Monomeric α-synuclein concentration is 100 µM. Proteinase K concentration is 3.8 µg/ml. Samples were withdrawn from the reaction, immediately after PK addition (lane most to the left, labelled 0) and at time 1, 5, 15, 30 and 60 min. PAGE analysis was performed and the gels were stained with Coomassie blue. The intensity of the proteolytic band generated at 15 min and indicated by the arrowhead was normalised to that of α-synuclein at time 0 for each PD (red) and MSA (green) patient sample ($n = 4$ PD and $n = 5$ MSA). PD patient-derived assemblies exhibited higher resistance to proteolysis than MSA patient-derived assemblies. In panel **c**, ***$P < 0.001$, two-sided unpaired Student's $t$ test. Source data for **a**, **c** are provided as a Source Data file.

the ones amplified with MSA homogenate, yielded profiles dominated by three bands. Quantification of the intensity of the band with the lowest molecular weight (arrowhead in Fig. 2c) that was common to the two profiles showed very significant differences between fibrils derived from PD and MSA cases. Comparison of the shapes and limited proteolytic profiles of the PD and MSA brain amplified and de novo assembled α-synuclein fibrils revealed several features. The fibrils amplified in the presence of PD or MSA brain homogenate were twisted while the de novo-generated polymorph fibrils were not. In addition, while both were twisted, the twist pitch of brain-amplified fibrils differed very significantly from that of the polymorph ribbons. Despite the differences in shape between brain-amplified assemblies and de novo-generated polymorphs fibrils and ribbons, the proteolytic pattern of fibrils derived from PD brains resembled that of the polymorph ribbons while that of fibrils amplified from MSA brains resembled that of the polymorph fibrils.

**Brain-amplified α-synuclein strains cause progressive degeneration of dopaminergic neurons.** Previous studies showed that sarkosyl-insoluble fibrils from human MSA brain or in vitro preformed fibrils propagated within oligodendrocytes transgenically expressing human α-synuclein have a higher aggregation propensity than LB-isolated α-synuclein fibrils[4]. However, these experiments were not performed in a human system and cannot entirely exclude potential confounding effects of post-translational modifications or co-purified proteins or lipids that are inherent to the isolation of insoluble material from the brain. To bypass this limitation, we investigated the effect of synthetic α-synuclein fibrils assembled de novo or amplified in the presence of PD ($n = 3$) or MSA ($n = 5$) brain homogenates. We tested these assemblies at two concentrations, 0.1 µM or 1 µM, using three clonal lines from SNCA$^{TRIP}$ or three different healthy control lines, each differentiated three times. We found that PD-amplified or MSA-amplified fibrils had similar seeding potency to de novo-generated fibrils at 2 weeks as measured by pSyn accumulation (Fig. 3a, b). Because each preparation had similar TEM characteristics (Fig. 2) and seeding propensity (Fig. 3a, b), we initially analysed blindly pooled PD or pooled MSA-derived fibrils on three SNCA$^{TRIP}$ clones and three HC clones, each differentiated three times. Despite similar seeding potency, when assessed morphologically, we observed that pSyn-positive aggregates in human neurons differed depending on the type of fibril used: specifically, MSA-amplified fibrils triggered the formation of short neuritic or speckled aggregates, PD-amplified fibrils

induced the formation of longer neuritic aggregates, whereas de novo-generated fibrils had an intermediate morphology (Fig. 3c). Aggregate morphology was quantified using Fiji ImageJ across three different clones of SNCA$^{TRIP}$ lines each differentiated three times as shown in Fig. 3d. A similar pattern was also detected in control lines (Supplementary Fig. 7) and was not related to the length of the seeds which was similar (60–100 nm) across all conditions (Supplementary Fig. 8). The different aggregate morphology induced by PD- and MSA-amplified fibrils corresponded to a distinct pattern of neuronal α-synuclein digestion after limited proteolysis with Proteinase K (Fig. 3e). Importantly, aggregated α-synuclein from PD-seeded neurons was more resistant to Proteinase K digestion than MSA-seeded neurons based on immunoblotting as detected after limited proteolysis of PD- vs MSA-amplified fibrils based on Coomassie blue staining (Fig. 2c and Supplementary Fig. 6).

We observed that in SNCA$^{TRIP}$ neurons, aggregates triggered by 1 µM of PD-amplified or MSA-amplified fibrils caused loss of neuronal axonal integrity as evidenced by TH (Fig. 3c) or TUJ1 immunostaining (Fig. 3f) and progressively increased nuclear fragmentation (Fig. 3g, h). Specifically, nuclear fragmentation was detected in ~30% of neurons in all conditions by 2 weeks but progressed by 3 weeks only in SNCA$^{TRIP}$ neurons seeded with PD-amplified fibrils (60%) and MSA-amplified fibrils (80%). Interestingly, this progressive phenotype was not observed in three different healthy control lines (Fig. 3g, h). Overall, the extent of neuronal death was higher in neuronal cultures seeded with MSA-amplified fibrils than PD-amplified fibrils (Fig. 3h). To exclude effects from contaminants, we tested blindly each brain-amplified batch against its corresponding brain homogenate that was used for amplification to its final dilution or recombinant monomer that was used as substrate 3 weeks after addition to SNCA$^{TRIP}$ neurons derived from one clone, differentiated three times. These experiments confirmed that cell death was caused by the amplified fibrils and not contaminants within the homogenate or monomeric α-synuclein (Fig. 3i). Collectively, these data show that in human neurons, brain-amplified fibrils generate distinct patterns of aggregation and proteotoxicity, supporting the existence of different conformers or strains within each disease that reflects severity in PD versus MSA.

**α-Synuclein monomer levels is a critical determinant of strain-induced toxicity in human neurons.** To investigate the molecular drivers of strain-induced neuronal loss, we assessed global transcriptomic profiles using bulk RNASeq in SNCA$^{TRIP}$ neurons at 1 and 2 weeks after seeding with 1 µM of de novo-generated

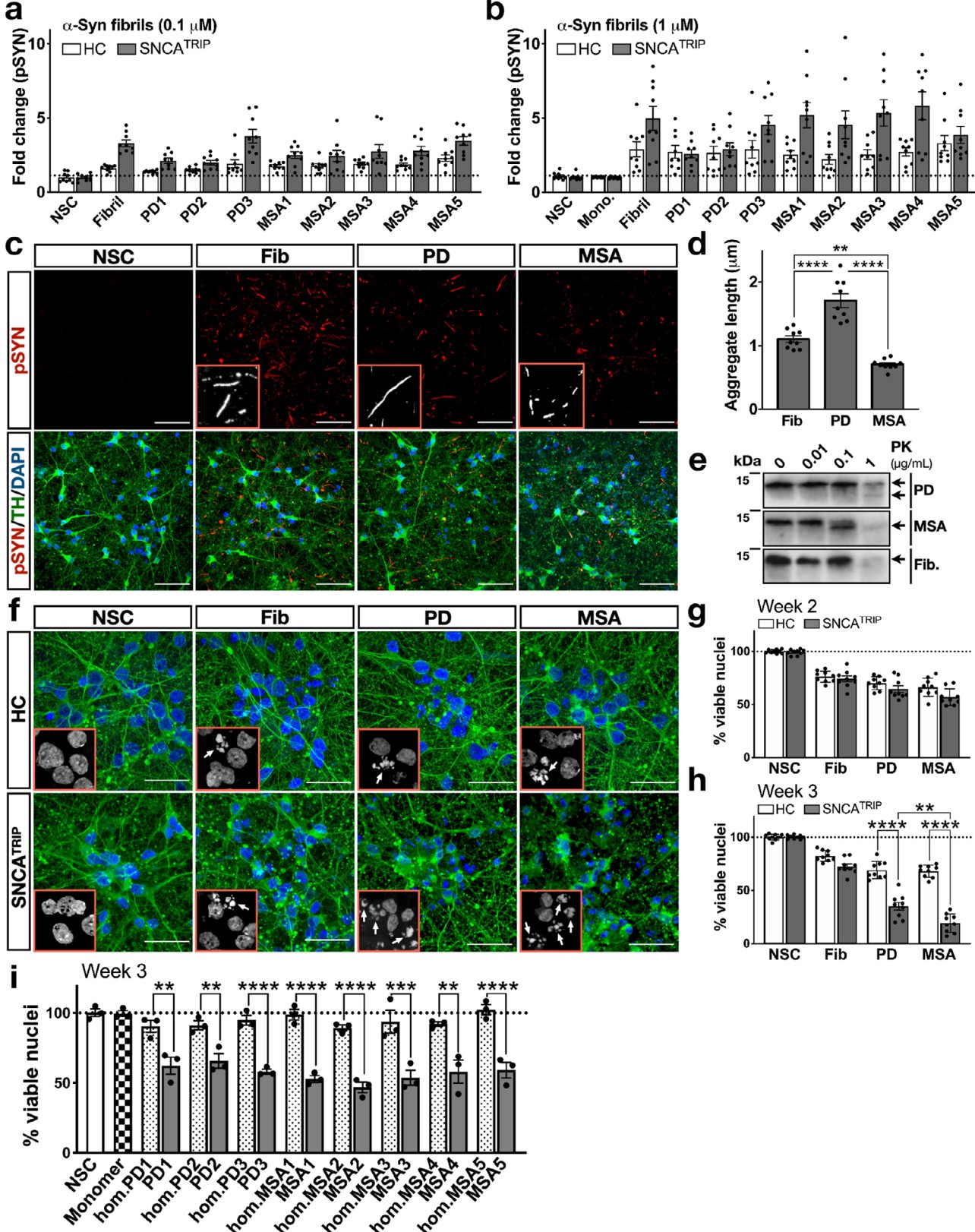

fibrils, pooled PD-amplified or pooled MSA-amplified fibrils. We chose week 1 post seeding because it was the earliest timepoint we observed consistent signal by FRET signifying endogenous α-synuclein aggregation (Fig. 1e), and week 2 post seeding because aggregation was more prominent while the neuronal loss was

minimal and similar across conditions (Fig. 3g). Principal component analysis (PCA) on the 1000 most variable protein-coding genes showed that samples separated based on time and clustered based on the presence of aggregation irrespective of the strain used to induce it (Fig. 4a). Based on this finding, we pooled all the

**Fig. 3 Aggregation propensity and neurotoxicity of synthetic and brain-amplified α-synuclein fibrils.** Similar levels of pSYN were detected by HTRF in neuronal lysates seeded with **a** 0.1 μM or **b** 1 μM fibrils when comparing de novo fibrils to each case of PD- or MSA-amplified strains ($n = 9$ for panels **a** and **b**). **c** Confocal images of SNCA$^{TRIP}$ dopaminergic neurons treated with de novo or brain-amplified fibrils. pSYN (upper), tyrosine hydroxylase (TH, lower). Images are representative of three independent differentiations (scale bar: 50 μm) and quantified in **d**, pSYN-positive aggregates induced by MSA-amplified fibrils were shorter and more speckled, whereas pSYN-positive aggregates induced by PD-amplified fibrils were neuritic and longer in length compared to ones induced by de novo-generated fibrils ($n = 9$). **e** Proteinase K digestion of cell lysates followed by immunoblotting with a mixture of anti-α-synuclein antibodies (ASyM, 4B12 and 10D2) revealed different fragments and resistance to proteolysis in PD-seeded versus MSA-seeded neuronal aggregates (immunoblot representative of $n = 2$). **f** Representative confocal images from three independent differentiations depicting nuclear fragmentation (DAPI) and axonal degeneration (TUJ1), which were especially prominent in SNCA$^{TRIP}$ neurons treated with PD- or MSA-amplified fibrils. Healthy control (upper) and SNCA$^{TRIP}$ (lower) lines were treated with de novo fibrils, PD- or MSA-amplified fibrils (scale bar: 30 μm). **g** Viability was similar across conditions at 2 weeks post seeding ($n = 9$). **h** Increased nuclear fragmentation was detected in SNCA$^{TRIP}$ lines seeded with MSA-amplified fibrils followed by PD-amplified fibrils at 3 weeks post seeding ($n = 9$). Healthy control lines did not exhibit a similarly progressive phenotype ($n = 9$). **i** α-Syn monomer and homogenates (hm) from each PD and MSA cases did not cause toxicity in contrast to corresponding amplified fibrils ($n = 3$). NSC = non-seeded control, which means untreated neurons, Mono = neurons treated with monomeric α-Syn. Each dot corresponds to one clone differentiated once and data are mean ± s.e.m from at least $n = 3$ differentiations per clone. In panels **d**, **g**, **h**, **i** *$P < 0.05$, **$P < 0.01$, ***$P < 0.001$, ****$P < 0.0001$ by one-way ANOVA followed by Tukey's multiple comparison test. Source data for **a**, **b**, **d**, **e**, **g**, **h**, **i** are provided as a Source Data file.

aggregation conditions and applied DESeq2 to identify differentially expressed genes between aggregation or baseline conditions at each timepoint. In total, 11 and 19 genes were significantly upregulated and 6 and 37 genes were downregulated in neurons with aggregates as compared to baseline at week 1 and week 2, respectively (Supplementary Fig. 9a). To identify potential cellular responses to α-synuclein aggregation, we performed Gene Set Enrichment Analysis (GSEA) using Gene Ontology (GO). Although PD-relevant pathways in organelle homoeostasis were enriched in the top 20 upregulated GO terms at week 1 and week 2 post seeding (Supplementary Fig. 9b), gene expression associated with these terms was not significantly affected by fibril addition.

We functionally assessed the impact of α-synuclein aggregation on one of these organelles, focusing on mitochondrial respiration. To this end, we performed extracellular flux analysis of oxygen consumption rates (OCR) at 2 weeks post seeding with 1 μM fibrils which triggered aggregation-induced cell death at 3 weeks. At 2 weeks post seeding with 1 μM of fibrils, there was a reduction in mitochondrial respiration which was similar across all strains when corrected for cell numbers in SNCA$^{TRIP}$ neurons (Fig. 4b). This deficit was similarly observed in healthy control neurons (Supplementary Fig. 10a) and was not associated with loss of mitochondrial mass as evident by Tom20 levels relative to actin or components of the respiratory complex (Fig. 4c and Supplementary Fig. 10b for quantification). Thus, although oxidative phosphorylation was reduced under conditions of α-synuclein accumulation and aggregation, it did not correlate with the extent of subsequent neuronal death seen with PD- or MSA-amplified α-synuclein fibrils. No differential deficit in oxidative phosphorylation was observed and no pathway was selectively identified by transcriptomics even at subtoxic levels of aggregation induced with 0.1 μM fibrils (Fig. 3a and Supplementary Fig. 11).

To investigate the relevance of *SNCA* expression and exclude genomic background confounders, we assessed the effect of PD- or MSA-amplified fibrils on a different pair of isogenic SNCA$^{TRIP}$ clones[11]. We found that isogenic correction of α-synuclein levels was sufficient to reduce aggregation (Fig. 4d) and reverse neuronal death (Fig. 4e). Therefore, a 2.5-fold increase in the levels of monomeric α-synuclein in SNCA$^{TRIP}$ neurons sufficiently explains their differential vulnerability, most probably due to the attainment of a critical burden of proteotoxic conformers.

**Toxicity of brain-amplified α-synuclein strains is potentiated by altered protein interactions.** Although a critical level of aggregation is necessary for neuronal loss, the distinct effects of brain-amplified and de novo-generated assemblies in SNCA$^{TRIP}$

neurons suggest additional mechanisms by which brain-amplified strains confer toxicity. One explanation is that toxicity may arise from differential aberrant strain interactions with the intracellular proteome, either by disrupting critical protein functions or evading homoeostatic defences. To investigate such mechanisms in the context of an intracellular α-synuclein assembly, we used proximity-dependent biotin identification, BioID[16]. To this end, we generated stable HEK293 cells expressing wild-type α-synuclein fused to the promiscuous mutant *Escherichia coli* biotin ligase (BirA*), in order to label interacting proteins within ~10 nm radius of α-synuclein aggregation induced with the addition of de novo-generated, pooled PD-amplified fibrils or pooled MSA-amplified fibrils (Fig. 5a). For these experiments, we employed a ratiometric BioID2 strategy using SILAC (Supplementary Fig. 12) and 3.1 μM biotin over 48 h to capture "historic" interactions over the course of the intracellular assembly of α-synuclein into aggregates induced by different strains. In control experiments, we confirmed that biotinylated proteins co-localised with α-synuclein-BirA* aggregates that were also phosphorylated at Ser129 (Fig. 5a). Affinity-purified biotinylated proteins were subjected to mass spectrometry.

Clustering analysis of aggregate-associated proteome under different conditions identified 56 differentially interacting proteins (Supplementary Data 1) when comparing de novo-generated to brain-amplified fibrils, including the PD-associated protein DJ-1 (Fig. 5b). Label-free quantification of the mass spectrometry data revealed the lower abundance of DJ-1 when the α-synuclein assembly was templated with pooled PD- or pooled MSA-amplified fibrils compared to de novo fibrils. Reduced biotinylation of DJ-1 in cells where the assembly was triggered with brain-amplified α-synuclein strains was confirmed by immunoprecipitation of DJ-1 and immunoblotting with streptavidin (Fig. 5c). DJ-1 is a deglycase[17] and loss of function mutations in this gene cause autosomal recessive PD[18] with α-synuclein pathology[19]. We, therefore, asked whether CRISPR/Cas9-mediated knockout of either endogenous DJ-1 or the related glyoxalase-1 (Glo-1) which act in the same pathway (Fig. 5d) impact on fibril-induced aggregation. For these experiments, we generated a clonal line stably expressing Cas9 and α-synuclein fused at the C-terminus with Venus. These experiments showed that knockout of either DJ-1 (Fig. 5e–g) or Glo-1 (Fig. 5h–j) increased aggregation seeded by de novo-generated or brain amplified fibrils and cellular dysfunction as measured by lactate dehydrogenase (LDH) release in conditioned media. Seeded aggregation was also associated with glycation of α-synuclein (Fig. 5k). Glycation involves the non-enzymatic modification of lysine, cysteine or arginine residues. α-Synuclein protein sequence

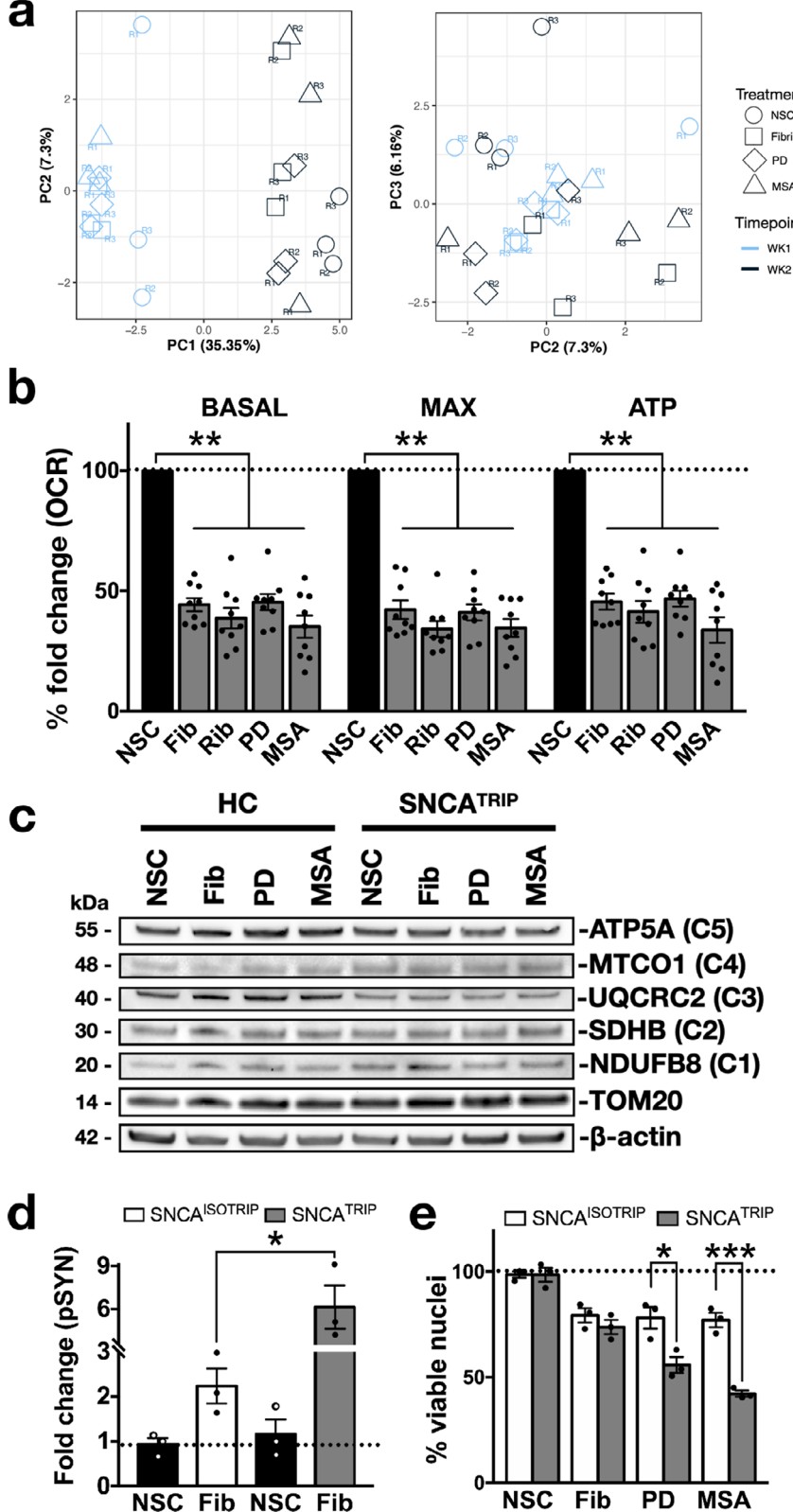

contains 15 lysine (K), but no cysteine or arginine residues. The N-terminus lysine residues of non-fibrillar α-synuclein from K6 to K45 were shown to be glycated in brain extracts[20] and we previously identified K12, K21, K45, K58 and K96 as sites of monomeric α-synuclein ubiquitination and degradation[21]. We, therefore, searched our mass spectrometry data for differential glycine–glycine–lysine-modified α-synuclein peptides, signifying differential ubiquitination. We identified glycine–glycine–lysine-modified peptide at K12 at much higher abundance in aggregates induced by de novo-generated fibrils compared to those induced by PD- or MSA-amplified fibrils (Fig. 5l and Supplementary Fig. 13). These data suggest that glycation may block specific

**Fig. 4 α-Synuclein expression is a critical determinant of strain-induced toxicity. a** Principle component analysis (PCA) on the 1000 most variable protein-coding genes showed that samples clustered by time (week 1 vs week 2) and treatment (aggregation versus baseline) but not by strain used ($n = 3$ per condition per timepoint). **b** De novo fibrils, ribbons, or brain-amplified fibrils similarly reduced OCR, maximal respiration and ATP production measured 2 weeks post seeding ($n = 9$). **c** No difference in the levels of TOM20 or respiratory complex components normalised to β-actin were detected in neuronal lysates 2 weeks post seeding. Representative immunoblots from three independent differentiations quantified as shown in Supplementary Fig. 10b. **d** Nuclear fragmentation in SNCA$^{TRIP}$ neurons 3 weeks post seeding was rescued by isogenic correction of SNCA expression ($n = 3$). **e** Isogenic correction of SNCA expression neurons reduced pSYN levels compared to SNCA$^{TRIP}$ 2 weeks post seeding as quantified by HTRF ($n = 3$). NSC = non-seeded control, which means untreated neurons. Each dot corresponds to one clone differentiated once and data are mean ± s.e.m from at least $n = 3$ differentiations per clone. In panels **b**, **d**, **e**, *$P < 0.05$, **$P < 0.01$, ***$P < 0.001$, by one-way ANOVA followed by Tukey's multiple comparison test. Source data for **b**, **c**, **d**, **e** are provided as a Source Data file.

lysine ubiquitination, potentially increasing the amount of aggregated α-synuclein within cells by reducing degradation. One prediction from these findings in HEK293 cells is that saturating the protective effects of DJ-1 with exogenous methylglyoxal (MGO) could potentiate aggregation and toxicity in iPSC-derived neurons seeded with de novo-generated fibrils. Accordingly, we found that two treatments of iPSC-derived dopaminergic neurons with 250 μM of MGO 3 days apart, given 2 weeks after seeding with fibrils, increased both aggregation (Fig. 5m) and aggregate-induced neuronal death (Fig. 5n).

To further validate these findings, we generated DJ-1 knockout iPSC clones in our SNCA$^{TRIP}$ line as detailed in the "Methods" and shown in Fig. 6a, b and Supplementary Figs. 14 and 15. These iPSC clones successfully differentiated into dopaminergic neurons to the same extent as the SNCA$^{TRIP}$ line of origin (Fig. 6c, d). At the baseline, α-synuclein levels were similar at DIV30 (Supplementary Fig. 16) but increased in DJ-1 knockout clones with time in culture (DIV60) compared to the SNCA$^{TRIP}$ line of origin (Fig. 6e, f). This was associated with a reduction in oxidative phosphorylation compared to healthy controls or SNCA$^{TRIP}$ neurons at DIV60 (Fig. 6g). Importantly, upon seeding with de novo or brain-amplified strains, dopaminergic neurons lacking DJ-1 exhibited accelerated cell death irrespective of the seed used that was already detected at week 1 post seeding (Fig. 6h–j) and higher aggregation in surviving neurons (Fig. 6k) as we found in cell lines (Fig. 5e–g). Collectively, these data suggest that differential interactions of α-synuclein strains with the cellular proteome as exemplified here by DJ-1 may influence their pathogenic effects whereas prevention of such interactions (e.g. by CRISPR/Cas9 knockout) impairs α-synuclein homoeostasis and increases aggregate-induced neuronal death.

## Discussion

We showed that α-synuclein assemblies amplified in the presence of human brain homogenate exert distinct patterns of intra-neuronal aggregation and protein interaction causing the death of human neurons with increased α-synuclein levels in a manner that reflects disease severity in patients with PD and MSA[22]. Taken together with recent animal studies[7], our data in human neurons suggest that disease severity in patients may arise principally from the differential toxicity of α-synuclein conformers or strains. This conclusion is supported by our analysis by TEM and limited proteolysis of the amplified fibrils and is in agreement with other studies showing that amplified fibrils generated with PD brain homogenate differ from fibrils generated with MSA counterparts or fibrils generated de novo[23,24]. The full extent to which fibrils amplified with brain homogenate resemble the ones found in the brain requires the resolution of their cryo-EM structures. Recently, the cryo-EM structure of fibrils amplified in the presence of sarkosyl-extracted fractions from MSA brain homogenates, using a method different from the one used in this study, were found to exhibit some degree of heterogeneity and differ from the twisted fibrils in the sarkosyl-extracted fractions[25].

This result may be due to the removal by the sarkosyl of relevant factors in the brain homogenate that are required for faithful amplification.

PMCA is an established methodology[23,24,26,27] that offers a means to investigate phenotypes that arise solely from different conformations as their amplification is devoid of contaminants or post-translational modifications that are inherent to studies on tissue extracts. In this context, an important finding of our work in pathophysiologically relevant iPSC-derived dopaminergic neurons and their isogenic controls is that vulnerability to neuronal death is primarily dependent on the level of α-synuclein expression, which in turn determines the critical burden of specific conformers that are generated inside cells. This concept is supported by studies in rodents where seeding and propagation along neuronal connections were also shown to involve neurons with a higher level of α-synuclein expression[28–30]. Interestingly, we found that seeded aggregation in human dopaminergic neurons led to an upregulation of SNCA mRNA expression which was most prominent in iPSC neurons with SNCA triplication. Although the molecular basis of this finding remains unknown and may involve epigenetic[31] or miRNA[32] dysregulation, it suggests that neurons with the highest level of SNCA expression are prone to intracellular amplification of aggregates due to feedback upregulation of α-synuclein monomer. Increased SNCA-mRNA has been reported in the midbrain[33] and in individually microdissected dopaminergic neurons of the substantia nigra from PD brains[34]. In addition, analysis of SNCA mRNA in the human temporal cortex in PD and dementia with LB also demonstrated a correlation between the number of α-synuclein immunoreactive LBs and the abundance of SNCA-mRNA expression[35].

How PD- or MSA-amplified assemblies cause neuronal death is not fully resolved and requires further studies. Our transcriptomic analysis identified GO terms involving ER stress and endosomal transport defects early in the aggregation cascade followed by impaired organelle fusion, mitochondrial membrane defects and apoptosis. Although gene expression associated with these terms was not significant, this analysis suggests that the toxic gain of function of aggregated α-synuclein is multifaceted and at least partly mediated by effects on organelle membrane function[36]. For example, mitochondrial impairment in PD has been ascribed to effects of α-synuclein oligomers on respiratory complex assembly and oxidative phosphorylation[37–39] or aggregate-induced mitochondrial fragmentation[15]. In agreement with such observations, we found reduced basal and maximal respiration in both SNCA$^{TRIP}$ and healthy control neurons upon seeded aggregation. This reduction in mitochondrial respiration was not associated in our experiments with decreased mitochondrial mass or components of the respiratory complexes suggesting a functional deficit that did not per se correlate with the extent of subsequent neuronal death.

Another important finding of our work is that strain-induced toxicity may arise from altered interactions with the cellular

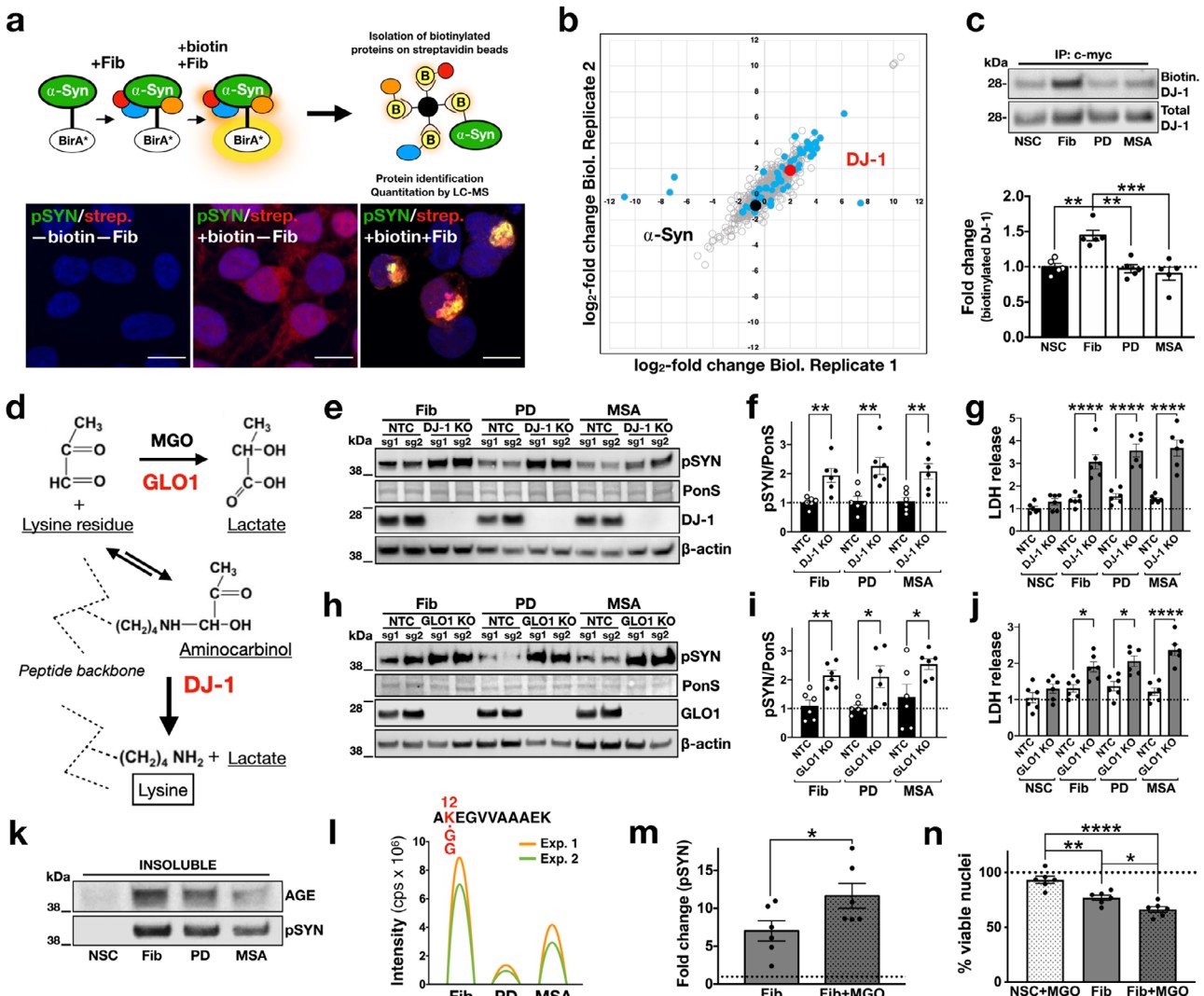

**Fig. 5 Differential interactome of synthetic versus brain-amplified strains. a** α-Synuclein-BirA* expressing clonal cells were exposed to fibrils and biotin, resulting in biotinylation of proteins in proximity to aggregating α-synuclein inside cells. Biotinylated proteins were captured using streptavidin beads. Cells were treated with fibrils and biotin and stained for pSyn (green) and streptavidin (red) to determine biotinylated proteins. Co-localisation indicates that aggregates contain biotinylated proteins. Images are representative of three independent experiments (scale bar: 10 μm). **b** Mass spectrometry of biotinylated proteins (1002 identified in total shown as circles) from cells seeded with de novo-generated or brain-amplified fibrils identified 56 differentially interacting proteins (blue dots), including DJ-1 (in red); SNCA is shown for comparison (black dot). **c** DJ-1 immunocapture followed by streptavidin immunoblotting confirmed the enhanced interaction between DJ-1 and intracellular α-synuclein assemblies seeded with de novo-generated fibrils compared to PD- or MSA-amplified fibrils ($n = 5$). **d** Pathway of MGO detoxification involving Glo-1 and DJ-1. **e, f** CRISPR/Cas9 knockout of the deglycase DJ-1, increased Ser129 phosphorylation of α-synuclein-venus (pSYN) induced by de novo fibrils, PD- or MSA-amplified fibrils compared to the non-target control and **g**, aggregate-induced toxicity as measured by LDH release ($n = 3$ per sgRNA). **h, i** CRISPR/Cas9 knockout of the glyoxalase Glo1, similarly increased Ser129 phosphorylation of α-synuclein-venus (pSYN) and **j**, aggregate-induced toxicity as measured by LDH release ($n = 3$ per sgRNA). Two sgRNAs (sg1, sg2) were used per target for the knockout experiments (total $n = 6$ per target). **k** pSyn-positive aggregated α-synuclein was immunoreactive for advanced glycation end-products. Immunoblot is representative of two independent experiments. **l** Illustrated chromatograms for each monoisotopic SILAC precursor ion that are shown in Supplementary Fig. 13. The highest abundance of GlyGly-modified K12 α-synuclein peptide was seen in de novo-generated fibril-treated samples with lower abundance in the PD- and MSA-amplified fibril-treated samples ($n = 2$). **m** MGO increased fibril-induced aggregation as measured by pSYN in iPSC-derived dopaminergic neurons ($n = 6$). **n** Exposure to MGO increased fibril-induced death of dopaminergic neurons as quantified by nuclear fragmentation ($n = 6$). NSC = non-seeded control, which means untreated cells. Each dot corresponds to one biological replicate (**c, f, g, i, j**) or one clone differentiated once (**m, n**) and data are mean ± s.e.m. *$P < 0.05$, **$P < 0.01$, by one-way ANOVA followed by Tukey's multiple comparison test (**c, f, g, i, j, n**) or two-sided unpaired Student's $t$ test (**m**). Source data for **c, e, f, g, h, i, j, k, m, n** are provided as a Source Data file.

proteome as shown for DJ-1. DJ-1 is a redox-sensitive chaperone that inhibits α-synuclein aggregation in vitro[40,41] and interacts with α-synuclein in cells[42]. Our biotinylation experiments showed that DJ-1 interaction with intracellular α-synuclein assemblies is strongest when seeded with de novo-generated fibrils, suggesting

that conformation may determine the strength of such interactions. Loss-of-function mutations in DJ-1 cause autosomal recessive PD with LB[19] and DJ-1 knockout in our experiments increased seeded aggregation and aggregate-induced neuronal loss. Beyond its chaperone activity, DJ-1 interaction with α-

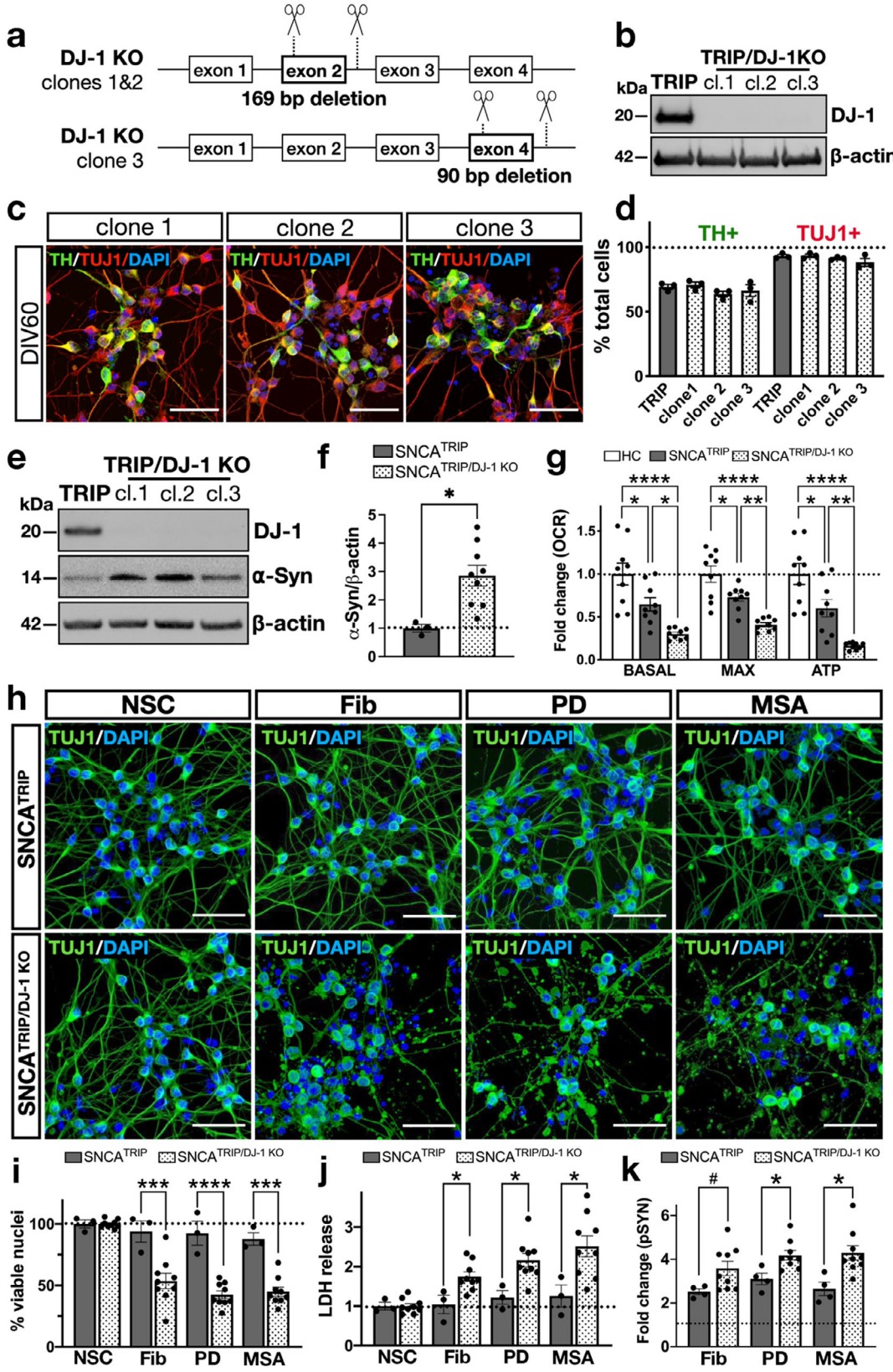

synuclein or other proximal proteins during the process of aggregation may serve a protective effect via deglycation[17]. Numerous epidemiological studies showed that pre-existing diabetes mellitus or dietary habits featuring high glycaemic carbohydrates increase the risk of developing PD[43–45], suggesting that enhanced glycation contributes to the pathogenic cascade. We found that aggregated α-synuclein was glycated in cells and knockout of endogenous glyoxalase increased aggregation. How

**Fig. 6 DJ-1 knockout in iPSC-derived dopaminergic neurons increased aggregate-induced toxicity. a** Approach used to knock out DJ-1 in the SNCA$^{TRIP}$ line. **b** Loss of DJ-1 protein in selected iPSC clones. Immunoblot is representative of n = 3 biological replicates. **c** SNCA$^{TRIP}$/DJ-1KO clones differentiated into dopaminergic neurons. Images are representative of three independent differentiations (scale bar: 30 μm). **d** Quantification of the percentage of TH- and TUJ1-positive neurons in DJ-1 knockout clones was similar to the corresponding SNCA$^{TRIP}$ iPSC line of origin (n = 3). **e** Immunoblotting at DIV60 showed increased α-synuclein levels at baseline in SNCA$^{TRIP}$/DJ-1KO neurons compared to the SNCA$^{TRIP}$ neurons derived from the iPSC line of origin and quantified in panel **f** (n = 9 for SNCA$^{TRIP}$/DJ-1KO and n = 3 for SNCA$^{TRIP}$). **g** Reduced OCR, maximal respiration and ATP production in SNCA$^{TRIP}$/DJ-1KO neurons at baseline DIV60 relative to the SNCA$^{TRIP}$ neurons and healthy control lines (n = 9). **h** Accelerated neuronal loss was detected already at 1 week post seeding in SNCA$^{TRIP}$/DJ-1KO clones irrespective of the strain used when compared to seeded neurons from the SNCA$^{TRIP}$ iPSC line of origin. Images are representative of three independent differentiations (scale bar: 30 μm). **i** Quantification of nuclear fragmentation in the SNCA$^{TRIP}$/DJ-1KO clones seeded with de novo-generated fibrils, PD-amplified fibrils or MSA-amplified fibrils at 1 week post seeding (n = 9 for SNCA$^{TRIP}$/DJ-1KO and n = 3 for SNCA$^{TRIP}$). At this timepoint, neurons from the SNCA$^{TRIP}$ iPSC line of origin did not exhibit a similarly severe phenotype. **j** Increased LDH release was detected in seeded SNCA$^{TRIP}$/DJ-1KO clones (n = 9 for SNCA$^{TRIP}$/DJ-1KO and n = 3 for SNCA$^{TRIP}$). **k** Despite the progressive cell loss, enhanced pSyn was detected in surviving SNCA$^{TRIP}$/DJ-1KO neurons at 2 weeks post seeding as quantified by HTRF (n = 9 for SNCA$^{TRIP}$/DJ-1KO and n = 4 for SNCA$^{TRIP}$). NSC = non-seeded control, which means untreated neurons. Each dot corresponds to one clone differentiated once and data are mean ± s.e.m from at least n = 3 differentiations per clone. #P = 0.06, *P < 0.05, **P < 0.01, ***P < 0.001, ****P < 0.0001 by one-way ANOVA followed by Tukey's multiple comparison test (**g**, **i**) or two-sided unpaired Student's t test (**f**, **j**, **k**). Source data for **b**, **d**, **e**, **f**, **g**, **l**, **j**, **k** are provided as a Source Data file.

glycation promotes neurodegeneration is unclear and likely multifaceted. Although glycation was detected in all conditions, our mass spectrometry analysis suggests that modification of specific lysine residues such as K12 on α-synuclein may prevent conjugation to ubiquitin[21]. This could potentially impair degradation as suggested previously by a study that identified K12 as a glycated residue[20]. This model is also compatible with the structure of recombinant or brain isolated fibrils where K12 is not involved in the amyloid core but rather exposed to the solvent[46,47] and subjected to ubiquitination[47]. Accordingly, saturating the deglycase or glyoxalase pathways with two pulses of MGO or DJ-1 knockout increased fibril-induced aggregation and neuronal death in our iPSC-based model. More broadly, glycation of assembled α-synuclein or other proximal proteins could overwhelm the glyoxalase pathway, which employs glutathione and NADPH to deactivate MGO, depleting antioxidant defences or cofactors necessary for their restoration. Therefore, by evading potentially protective interactions as exemplified here by DJ-1, assembled α-synuclein may acquire or induce modifications that enhance its neurotoxic effects.

Increasing evidence indicates that PD and MSA are associated with distinct α-synuclein strains. Our data reinforce this notion and suggest that monomeric α-synuclein which determines the abundance of generated assemblies and conformation which dictates their protein interactions, influence their pathogenicity. Therefore, strategies aimed at activating DJ-1 or reducing α-synuclein would be promising targets against strain-induced dopaminergic neuronal death. More broadly, our model represents a unique human system to interrogate modifiers of neuronal loss that is triggered by intraneuronal accumulation of brain amplified α-synuclein polymorphs.

## Methods

**Generation of iPSC-derived midbrain dopaminergic neurons.** Dopaminergic neurons were derived from human iPSC lines[12]. iPSCs were seeded onto Geltrex-coated six-well plates, expanded until >80% confluency and mTeSR1 media was changed to day in vitro (DIV) 0 media (2 μM A83-01, 100 nM LDN in neural induction base medium). DIV1-4 media contained 2 μM A83-01, 100 nM LDN, 300 ng/ml SHH C25II, 2 μM purmorphamine and 200 ng/ml FGF8a in neural induction base medium. In total, 3 μM CHIR-99021 was added from day 3 until day 12. DIV5-6 media contained 100 nM LDN, 300 ng/ml SHH C25II, 2 μM purmorphamine, 3 μM CHIR-99021 and 200 ng/ml FGF8a in neural induction base medium. DIV7-10 media contained 100 nM LDN and 3 μM CHIR-99021 in neural induction base medium. DIV11-19 medium contained 20 ng/ml BDNF, 20 ng/ml GDNF, 1 ng/ml TGFβ3, 10 μM DAPT, 200 μM ascorbic acid, 500 μM db-cAMP in neural differentiation media and 1 μg/ml of laminin was added from 17 to 25 days of differentiation. At DIV20, cells were replated onto geltrex-coated plates, or coverslips. From DIV21, media changes were done with DIV11-19 components until analysis timepoints were reached.

**Addition of exogenous fibrils to cells.** Dopaminergic neurons were treated with either de novo-generated polymorphs (fibrils or ribbons) or brain-amplified fibrils (PD: three cases, MSA: five cases) on DIV45 and assayed at weekly timepoints from 1 to 4 weeks post seeding. Fibrils were diluted in differentiation media DIV11-19 and added directly to neurons in varying concentrations.

**Immunofluorescence staining.** iPSC-derived neurons were fixed in 4% paraformaldehyde (PFA), permeabilised in 0.1% Triton-X-100 followed by blocking in 3% BSA, 3% goat serum containing PBST (0.1% Tween-20) and then incubation with primary antibodies and fluorescently labelled secondary antibodies. The following primary antibodies were used: anti-phospho S129 α-synuclein (rabbit monoclonal, Abcam #ab51253 (EP1536Y); 1:1000 dilution), anti-tyrosine hydroxylase clone LNC1 (mouse monoclonal, Millipore #MAB318; 1:250 dilution), anti-βIII tubulin/TUJ1 (mouse monoclonal, BioLegend #MMS-435P; 1:1500 dilution), anti-MAP2 (chicken polyclonal, Abcam #ab5392, 1:2000 dilution). The following secondary antibodies were used: Alexa Fluor 594-labelled goat anti-rabbit IgG H&L (Abcam #150080, 1:1500 dilution), Alexa Fluor 488-labelled goat anti-mouse IgG H&L (Abcam #150113; 1:1500 dilution), Alexa Fluor 488-labelled goat anti-chicken IgY H&L (Abcam #150169; 1:1500 dilution). Images were obtained using Zeiss LSM 710 confocal microscope (Carl Zeiss AG, Oberkochen, Germany). For dopaminergic markers, ten images (×40 magnification) were analysed for each marker per differentiation per line. To measure aggregate length, a single image consisting of between 500 and 2000 events (×63) was obtained per timepoint per clone per differentiation and analysed using the FIJI ridge detection plugin (v1.4.0). For nuclear morphology at least five images (×40 magnification) were obtained per timepoint per differentiation using immunofluorescence microscopy and in total >200 cells were counted per clone per condition for nuclear fragmentation measurements. Nuclear fragmentation counts were blinded and quantified manually.

**Time-resolved FRET.** Anti-oligomeric and anti-phospho S129 α-synuclein kits were purchased from Cisbio (Perkin Elmer, Waltham, MA USA) and used according to the manufacturer's instructions, including optimisation experiments to determine the dynamic range of the assay. Donor and acceptor FRET fluorophores were mixed with cell lysate and incubated for either 24 h (oligomeric α-synuclein kit) or 1 h (phospho S129 α-synuclein kit). Samples were excited at 340 nm and emission values were measured at 620 nm and 665 nm using a CLARIOstar Plus (BMG Labtech, Ortenberg, Germany). FRET was determined by calculating the ratio, 665/620 nm, with α-synuclein aggregation being detected as a background-subtracted fold increase over non-seeded controls (%ΔF).

**Relative gene expression.** The total RNA was isolated using the RNeasy Kit (Qiagen, Hilden Germany) according to the manufacturer's instructions. In total, 1 μg of cDNA was synthesised with random hexamers using the QuantiTect Reverse Transcription Kit (Qiagen, Hilden Germany). To quantify relative SNCA mRNA levels, real-time PCR was performed by combining in each reaction 5 ng of cDNA, LightCycler 480 SYBR Green I Master Mix (Roche Life Science, Penzberg, Germany) and 1 μM of both forward and reverse primers. Data were normalised to two housekeeping genes, GAPDH and ACTB. All primer sequences are shown in Supplementary Table 3. Samples were analysed using Roche LightCycler 480 Instrument (Roche Life Science, Penzberg, Germany) and its associated software was used to calculate crossing threshold (Ct) values. Relative gene expression was subsequently calculated using the ΔΔCt method.

**Metabolic studies.** Oxygen consumption (OCR) rates were measured using the Seahorse XFe96 analyser (Seahorse Bioscience) in cells seeded on XFe96 microplates. The Mito stress test kit (Seahorse Bioscience) was used to monitor OCR.

Three baseline recordings were made, followed by sequential injection of the ATP synthase inhibitor oligomycin (3 μM), the mitochondrial uncoupler carbonyl cyanide-4-(trifluoromethoxy)phenyl-hydrazone (FCCP; 1.25 μM), and the respiratory chain inhibitors antimycin A (0.5 μM) and rotenone (0.5 μM). Results were corrected for cell numbers using CyQuant (Thermo Fisher Scientific).

**Cell culture and conditions**. HEK293 cells (ATCC CRL-1573) were cultured in complete DMEM supplemented with 10% FBS and 100 μg/ml penicillin and 100 μg/ml streptomycin, at 37 °C and 5% $CO_2$. For the generation of single-cell clones, HEK293 cells were transiently transfected with α-Syn-BirA* and were selected with 300 μg/ml G418 (Gibco). For SILAC labelling of the single-cell clone, stably expressing WT α-Syn-BirA*, cells were cultured for about ten doublings in either media containing either R0K0, R6K4 or R10K8 media (Gemini Biosciences), supplemented with 10% dialysed FBS (10k MW cut-off, DC Biosciences), 100 μg/ml penicillin and 100 μg/ml streptomycin. For sgRNA knockdown studies, a clonal HEK cell line stably expressing WT α-Syn-venus and Cas9 was used. For fractionation of HEK293 cells, lysates were spun down at $21,000 \times g$. The supernatant was kept as the RIPA soluble fraction and the pellet was resuspended in 2% SDS buffer and subjected to sonication to re-solubilise the pellet. Samples were spun down at $21,000 \times g$ and the supernatant containing the insoluble fraction was used for western blotting to determine levels of pSyn.

**CRISPR/Cas9 modification of iPSC lines**. The generation of the isogenic line for SNCA Triplication was described in detail previously[11]. In brief, Cas9 nickase sgRNAs were designed to target SNCA exon 4 and applied to the ND34391G iPSC line (obtained from the Coriell Institute, distributed through the NINDS Fibroblasts and iPSCs Collection). After the clonal expansion of Cas9 nickase-transfected iPSCs, four clones were identified as variants by high-resolution melt analysis and one clone was confirmed by sequencing as a double knockout. The double-knockout clone (i.e. deletion of the two SNCA gene copies) was sequenced for off-target effects and quality controlled[11]. α-Synuclein mRNA and protein levels in the isogenic iPSC line were tested and shown to be reduced to levels comparable to the healthy control iPSC line[11]. In this context, isogenic refers to the fact that the parental SNCA triplication iPSC line and the double-knockout clone possess the identical genetic background, which includes overexpression of the other genes in the triplication locus, and selective reduction of SNCA gene dosage to achieve normal expression levels.

To knock out DJ-1, the SNCA^TRIP iPSC clone SFC831-03-03 was nucleofected with the CRISPR complexes formed by the two sgRNAs targeting either exon 2 or exon 4 and the HiFi Cas9 protein. For exon 2, two sgRNAs were designed targeting exon 2 and the downstream intron in order to remove a part of the gene causing deletion of 169 base pairs. For exon 4, two sgRNAs were designed targeting exon 4 and the downstream intron causing deletion of 90 base pairs. Single-guide RNAs for genome editing were dissolved in DNAse- and RNAse-free buffer to a concentration of 100 mM. One microliter of each sgRNA was mixed with 2 μL of HiFi Cas9 protein (IDT, Cat # 1081067) to form the active CRISPR/Cas9 complex. iPSCs were cultured in six-well plates coated with Matrigel (Corning Bioscience) in E8 medium until they reached a density of 70–90%. iPS cells were detached using Accutase (Gibco) and a total of $1.5 \times 10^6$ cells were co-nucleofected with the CRISPR/Cas9 complex. For the nucleofection, the P3 Primary Cell Kit (Lonza) was used and run with the programme CA167. iPSCs were subsequently transferred back to a Matrigel-coated 100-mm dish in E8 medium supplemented with 1:200 diluted Revita cell supplement (Gibco). After 1 week, 96 colonies from each nucleofection were picked and analysed by PCR for either exon 2 or exon 4. The deletions were confirmed by two primer pairs, one pair placed around the deleted part and the other pair consisting of one primer within the deleted part and one outside of it. After single-cell plating, three clones (two with exon 2 deletion and one with exon 4 deletion) were expanded and confirmed by sequencing of the targeted region. Their identity in relation to the original SNCA^TRIP clone was also confirmed by a short tandem repeat analysis. Protein knockout was confirmed by immunoblotting. Primer and sgRNA sequences are shown in Supplementary Table 3.

**Affinity capture of biotinylated proteins**. A C-terminal BioID2 tagged α-Syn construct was generated by introducing full-length α-Syn cDNA (ENST00000336904.7) into the MCS-BioID2-HA vector cloning vector (Addgene) using the restriction enzyme sites NheI and Kpn2I (Thermo Fisher). Clonal lines expressing BioID tagged α-Syn (termed α-Syn-BirA*) were selected and tested for aggregation upon the addition of exogenous fibrils. Clonal cells were cultured in media containing R0K0 or R6K4 or R10K8 for at least ten doublings to fully incorporate the stable isotopes. SILAC-labelled HEK293 cells were plated in six-well plates at a concentration of $9.8 \times 10^5$/well. Forty-eight hours after plating, fibrils were added to SILAC-labelled cells. R0K0 cells were treated with de novo-generated fibrils, R6K4 cells were treated with PD-amplified fibrils and R10K8 cells were treated with MSA-amplified fibrils at a final concentration of 1 μM. Three hours after seeding, appropriate media containing biotin was added to all cells at a final concentration of 3.1 μM. After 24 h, media was replaced supplemented with biotin. A further 24 h later, cells were harvested in cell lysis buffer (50 mM Tris, pH 7.4, 500 mM NaCl, 0.4% SDS, 5 mM EDTA, 1 mM DTT, and 1× Complete protease

inhibitor (Roche)) or fixed in PFA for immunofluorescence. Triton-x was added to protein lysates to a final concentration of 2%. Cell lysates were sonicated for two cycles at 30% amplitude 1 s on, 1 s off for 30 s and centrifuged at $16,000 \times g$. The supernatant was stored, and the pellet was resuspended in lysis buffer containing 2% SDS and sonicated for one sonication cycle, before centrifugation at $16,000 \times g$. The supernatant was pooled with the previous lysate. Levels of total cellular protein were measured in the Pierce BCA Protein Assay Kit (Thermo Fisher). In all, 1 mg of total protein from each of the three different experimental treatment groups R0K0, R6K4 and R10K8 were combined and incubated overnight on 300 μl of Myone streptavidin C1 beads (Thermo Fisher) with constant rotation at 4 °C. Beads were collected using a magnetic separation stand and were washed sequentially with wash buffer 1 (2% SDS), wash buffer 2 (0.1% deoxycholic acid, 1% Triton-x, 1 mM EDTA, 500 mM NaCl, 50 mM HEPES pH 7.5) and finally wash buffer 3 (0.1% deoxycholic acid, 0.5% NP-40, 1 mM EDTA, 250 mM LiCl, 10 mM Tris.Cl pH 7.4). Beads were finally washed with 50 mM Tris.Cl pH 7.4 and resuspended with 1.5 ml of 50 mM Tris.Cl. In total, 15% of the resuspended beads were saved for further analysis by western blot, and the remaining 85% were used for mass spectrometry. Beads were spun for 5 min at 2000 × g for 5 min to collect beads, and the supernatant was removed.

**Mass spectrometry and data analysis**. Beads were spun down and resuspended in 150 μl trypsin buffer (Promega). Samples were reduced using 10 μl of DTT reducing agent (final concentration 10 mM) and incubated for 45 min at RT. Samples were then alkylated with iodoacetamide (final concentration 40 mM) and incubated for 45 min at RT. SMART Digest Trypsin (Thermo Fisher) was added in a 1:50 ratio with respect to total protein content and samples were digested at 70 °C for 1 h. Samples were analysed on a nLC-MS/MS system consisting of an Orbitrap Fusion Lumos and Dionex Ultimate 3000 (Thermo Fisher). Peptides were separated on an Easyspray column (50 cm) with a gradient of 2-35% ACN in 5% DMSO in 0.1% formic acid over 60 min. MS1 spectra were acquired with a resolution of 120 k and an AGC target of 4E5. MS2 spectra were acquired in the linear ion trap after HCD fragmentation @ 28% normalised collision energy for up to 35 ms and an AGC target of 4E3. Selected precursors masses were excluded for 60 s. Data analysis was performed in Maxquant (v1.6.2.3)[48] using SILAC settings (R0K0, R6K4 and R10K8). Peptide precursor ion abundance was used to compare the volume under the chromatographic signal of each peptide detected for a protein. Peptide signals were then summarised into ratios for each protein across the different conditions, yielding a fold-change value. Specifically, the ratio of MSA- or PD-amplified to de novo fibrils was first calculated and then converted to log2 followed by statistical analysis within the biological replicates to yield fold changes and multiple testing corrected P values. Carbamidomethylation (Cys) was a fixed modification and oxidation (Met), phosphorylation (Ser, Thr, Tyr) and protein N-terminal acetylation were set a variable. Searches were performed with a mass tolerance of 10 ppm for precursors and 0.5 Da for fragments. Swissprot/Trembl FASTA file was used as a database. Protein and peptide FDR were set to 1% with the Match between Runs option enabled. The mass spectrometry proteomics data have been deposited to the ProteomeXchange Consortium via the PRIDE partner repository with the dataset identifier PXD024198.

**Immunoprecipitation and immunoblotting**. To confirm the interaction between synuclein and DJ-1, HEK293 α-Syn-BirA cells overexpressing DJ-1-myc were treated with de novo-generated or brain-amplified fibrils. Cell lysates were harvested in 0.1% SDS RIPA and anti-myc immunoprecipitation was performed. Immunoblotting for dopaminergic or HEK293 cell lysates was performed using standard protocols after lysis in RIPA Extraction Buffer (Thermo Fisher Scientific) supplemented with NEM (Sigma), PMSF (Sigma) and protein and phosphatase inhibitors (Sigma). Samples were mixed with Laemmeli buffer and heated for 5 min at 95 °C and spun down before separated on 4–12% Bis-tris gels, and transferred to a nitrocellulose membrane. Membranes were fixed in 0.4% PFA for 20 min before blocking in 5% BSA solution. Subsequently, membranes were probed overnight with primary antibodies and secondary antibody in blocking buffer for 1 h at RT. Biotinylated proteins were detected with HRP-conjugated streptavidin. Blots were scanned using Chemidoc Imager (Bio-Rad, Hercules, CA, USA), and band intensity was quantified using ImageJ (v2.1.0/1.53c). The following primary antibodies were used for immunoblotting: anti-PARK7/DJ-1 (Abcam #18257, 1:5000 dilution), anti-GLO1 (Abcam #129124, 1:5000 dilution), anti-PSyn EP1536Y (Abcam #51253; 1:1000 dilution), Syn-1 (aka Clone42, BD Biosciences #610787; 1:1,000 dilution), anti-beta Actin (Abcam #8826; 1:10,000 dilution), anti-TOM20 (Santa Cruz #FL-145; 1:1000 dilution), total OXPHOS rodent antibody cocktail (Abcam #110413; 1:1000 dilution), anti-AGE (Advanced Glycated End-products antibody, Merck Millipore #AB9890; 1:1000 dilution). Anti-c-Myc (9E10, Santa Cruz Biotechnology #sc-40, 10 μg per reaction) was used for immunoprecipitation. The following secondary antibodies were used: HRP-labelled goat anti-rabbit IgG H&L (Thermo Fisher #31460; 1:5000 dilution), HRP-labelled goat anti-mouse IgG H&L (Thermo Fisher #31430, 1:5000 dilution), HRP-labelled rabbit anti-goat IgG H&L (Thermo Fisher #31402, 1:5000 dilution), DyLight 680-conjugate anti-mouse IgG H&L (Cell Signalling #5470, 1:10,000 dilution), DyLight 800 PEG-conjugate anti-rabbit IgG H&L (Cell Signalling #5151, 1:10,000 dilution).

**CRISPR/Cas9 knockdown in HEK cells**. Two sgRNAs per target (DJ-1 or GLO-1) were designed using the online tool (http://crispor.tefor.net/) and cloned into a lentivirus backbone. All sgRNA sequences are shown in Supplementary Table 3. To generate the virus, HEK293T cells were transfected with the plasmid. After 6 h, media was changed and harvested a further 48 h later. HEK293 WT α-Syn-venus Cas9 cells were plated at a density of $3.5 \times 10^5$/well. Forty-eight hours later, cells were transduced with 200 μl of conditioned viral media. Twenty-four hours later, media were replaced and a further 24 h later, cells were selected with 1.5 μg/ml puromycin for 3 days. Cells were then replated for immunofluorescence, immunoblotting or RNA extraction. Two days later, cells were seeded as described above and were harvested after further 48 h.

**Human substantia nigra transcriptomic cell-type atlas**. Filtered substantia nigra single nuclei count data with cell-type annotations for seven post-mortem individuals was downloaded from the Broad Institute Single Cell Portal [https://singlecell.broadinstitute.org/single_cell]. The count matrices for all individuals were merged to result in 40,453 SN single nuclei with 24 cell types encompassing all known resident cell classes as identified by Welch et al.[13]. For the purpose of the cell-type deconvolution analysis in this study, these cell types were merged into eight major subtypes: astrocytes, dopaminergic neurons, endothelial, excitatory and inhibitory neurons, microglia, oligodendrocytes and oligodendrocyte progenitor cells.

**Bulk RNA sequencing**. Bulk RNASeq was performed to assess response to aggregation induced by de novo- or brain-amplified strains under different experimental conditions: subtoxic aggregation using 0.1 μM seeds in SNCA$^{\text{TRIP}}$ or HC iPSC-derived neurons ($n = 24$) or at two different timepoints (week 1 and week 2) using 1 μM seeds in SNCA$^{\text{TRIP}}$ iPSC-derived neurons ($n = 24$). RNA samples were prepared using standard poly-A enrichment, 250–300 bp insert cDNA library was prepared with the NEB Next® Ultra™ RNA Library Prep Kit and sequenced on Illumina NovaSeq 6000 (S4) using 150 bp paired-end reads. Sequencing resulted in ~40 million reads per sample. RNAseq data are available at GEO under accession numbers GSE149632 and GSE171999.

**Cell-type deconvolution of bulk RNAseq samples**. Cell-type composition of the bulk RNAseq samples was characterised using MuSiC (v0.1.1)[14], a cross-subject method that utilises only the consistently expressed genes across multiple cells by up-weighting genes with low cross-subject variance (informative genes) and down-weighing genes with high cross-subject variance (non-informative genes). Additionally, we accounted for a potential bias based on differing cell-type numbers across the cell types in the reference single-cell atlas by randomly down-sampling the eight major resident cell types to the same number ($n = 50$) 100 times and then estimating the cell-type composition of the bulk samples and averaging the proportions across the bootstraps to obtain the final cell-type estimates.

**Analysis of RNAseq data**. Kallisto (v0.46.2)[49] was used to estimate abundance for Ensembl (release 88) protein-coding genes from RNAseq reads. Count data were imported into R (v4.0.3) using Tximport (v1.18.0) and genes with less than ten reads across the six samples used in the comparison were filtered out. Differential gene expression was performed in DESeq2 (v1.30.0)[50] using the Wald test with an FDR threshold of 0.05 and a log2 fold-change threshold of 0.25. Gene Set Enrichment Analysis (GSEA) was performed using the Fgsea package (v1.16.0)[51] in R (v4.0.3). Genes were ranked for GSEA using the Wald statistic and Fgsea (v1.16.0) was run with 1000 permutations. Principal component analysis and all plotting were performed in R (v4.0.3).

**α-Synuclein purification and assembly into fibrillar polymorphs**. Recombinant human wild-type α-synuclein was expressed in *E. coli* BL21 DE3 CodonPlus cells (Stratagene, San Diego, CA, USA), purified[52] and assembled for 7 days into the fibrillar polymorph "fibrils" and "ribbons"[5]. The nature of α-synuclein fibrillar polymorphs was assessed by transmission electron microscopy (TEM) after adsorption of the fibrils onto carbon-coated 200 mesh grids and negative staining with 1% uranyl acetate using a Jeol 1400 transmission electron microscope. The images were recorded with a Gatan Orius CCD camera (Gatan, Pleasanton, CA, USA). The resulting α-synuclein fibrillar polymorphs were fragmented by sonication for 20 min in 2-ml Eppendorf tubes in a Vial Tweeter powered by an ultrasonic processor UIS250v (250 W, 2.4 kHz; Hielscher Ultrasonic, Teltow, Germany) to generate fibrillar particles that are suitable for endocytosis, with an average size 60 nm, as assessed by TEM analysis. Their proteinase K proteolytic patterns were assessed by SDS-PAGE analysis after staining with Coomassie blue. To this end, de novo-generated α-synuclein fibrillar polymorph (1.4 mg/ml) in 150 mM KCl, 50 mM Tris- HCl, pH 7.5 were treated at 37 °C with Proteinase K (3.8 μg/ml) (Roche). Aliquots were removed at different time intervals following the addition of the protease and transferred into Eppendorf tubes maintained at 90 °C containing sample buffer (50 mM Tris-HCl, pH 6.8, 4% SDS, 2% beta-mercaptoethanol, 12% glycerol and 0.01% bromophenol blue) to arrest immediately the cleavage reaction. The tubes were incubated 5 min at 90 °C before SDS-PAGE (15%) analysis.

**Patient brain homogenate preparation and quantification of pathogenic α-synuclein**. Frozen brain tissues were weighed in 50-ml Falcon tubes prior to the addition of the PMCA buffer (150 mM KCl, 50 mM Tris-HCl pH 7.5) in order to obtain a homogenate at 20% (weight:volume). The homogenisation was performed by sonication using the SFX 150 Cell Disruptor sonicator with a 3.17-mm microtip probe (Branson) for 1 min, with 10 s pulses followed by 10 s pauses in a biosafety level 3 environment (BSL-3). The homogenates were aliquoted and immediately frozen in liquid nitrogen before storage at −80 °C. All contaminated surfaces were cleaned with SDS. Pathogenic α-synuclein was quantified using both a filter retardation assay and immunoblotting and a FRET assay. The brain homogenates (0.5 mg) were immobilised on cellulose acetate membranes (0.2-μm pore size, Millipore Corp., Bedford, MA) by filtration using a 48-slot slot-blot filtration apparatus (GE Healthcare). The membranes were blocked in 5% dried skimmed milk and probed with the anti-α-synuclein antibody 4B12 (Biolegend, cat # 807801) and anti-P-S129 α-synuclein antibody 81A (Millipore, cat# MABN826). Following proteinase K digestion, neuronal lysates were immunoblotted with a mixture of ASyM (clone number 4.2, Agrisera, AS13 2719, residues 1–15), 4B12 (Biolegend, cat # 807801, residues 103–108) and 10D2 (Merck cat # MABN633, residues 118–127) antibodies to cover well the α-synuclein primary structure. Following wash with TBST, the membranes were incubated with HRP-conjugated goat anti-mouse IgG3 secondary antibody (Thermo, cat # M32707) for 1 h at room temperature. Proteins were visualised using ECL reagents (Pierce, USA).

Pathogenic phosphorylated α-synuclein was also quantified using the Cisbio FRET assay (Cisbio, France, cat # 6FASYPEG) following the manufacturer's recommendations. Patients brains homogenates were diluted to 10% (weight:volume) in lysis buffer provided in the HTRF kit. In total, 16 μL of each diluted brain homogenate were loaded into a 96-wells plate and mixed with 4 μL of the FRET donor and acceptor antibodies in the kit. The plate was sealed with a film (CmlAB, Denmark, cat # 13076-9P-500) and incubated for 20 h at 20 °C without shaking in a Thermomixer comfort (Eppendorf, Montesson, France). After incubation, time-resolved FRET was measured upon excitation at 337 nm using a plate reader (CLARIOstar, BMG Labtech, Germany). The FRET signal was recorded at two different wavelengths (665 nm and 620 nm). The amount of pathogenic α-synuclein was derived from the 665/620 nm fluorescence ratio and multiplied by 10.000.

**Amplification of pathogenic α-synuclein in patient brain homogenates by protein misfolding cyclic amplification**. All operations were performed in BSL-3. Each patient brain homogenate (2%, weight:volume) in PMCA buffer (150 mM KCl, 50 mM Tris-HCl, pH 7.5) containing monomeric α-synuclein (100 μM) was split into two tubes of PCR strips (BIOplastics, Landgraaf, The Netherlands). PMCA amplification was performed in quadruplicates for each patient using the Q700 generator and a 431MPX horn (Qsonica, Fisher Scientific, Illkirch, France). The power of the horn was set to 30% of maximal amplitude. The amplification cycles consisted in 15 s of sonication and 5 min of elongation of the resulting assemblies at 31 °C. Every hour, 5 μl were withdrawn from each tube and diluted in 300 μl of 10 μM of Thioflavin T. The amplification was monitored by measuring Thioflavin T fluorescence using a Cary Eclipse Fluorescence Spectrophotometer (Agilent, Les Ulis, France) with fixed excitation and emission wavelength at 440 and 480 nm, respectively. The time at which an aliquot from one amplification reaction was withdrawn for a subsequent reaction was defined as the time where ThT fluorescence increases for amplification reactions performed with PD and MSA brain homogenates, and not in reactions containing control patients brain homogenates or monomeric α-synuclein alone. Cycle 2 and 3 were performed following the same protocol using 2% of the preceding cycle reaction as seeds for PD cases, 5% for MSA cases. The amounts of brain homogenates and PMCA-amplified assemblies used in each amplification reaction were defined through an optimisation study aimed at maintaining high stringency by minimising the de novo aggregation of α-synuclein under the same experimental conditions. The PMCA reaction products resulting from the third amplification cycle were spun for 30 min at 100,000 × g, the amount of monomeric α-synuclein in the supernatant was assessed spectrophotometrically, and the pelleted assemblies were resuspended in phosphate-buffered saline (PBS) buffer and their shape was assessed by TEM. All resuspended assemblies, at a final concentration of 100 μM α-synuclein, were fragmented by sonication for 20 min in 2-mL Eppendorf tubes in a Vial Tweeter powered by an ultrasonic processor UIS250v (250 W, 2.4 kHz; Hielscher Ultrasonic, Teltow, Germany), aliquoted, flash-frozen in liquid nitrogen and stored at −80 °C until they were added to cell cultures. Their size was measured in order to ascertain that their length (60–100 nm) was similar to that of de novo-generated fibrils and ribbons and their limited proteolytic fingerprints analysed as described for α-synuclein fibrillar polymorphs generated de novo.

**Statistical analyses**. All statistical analyses were performed using GraphPad Prism 7 (GraphPad Software Inc., La Jolla, CA, USA). All data were examined for normality, and statistical tests were chosen accordingly. For normally distributed data, one-way ANOVA was used. For non-normally distributed data, the Kruskal–Wallis test was used. A two-sided unpaired Student's *t* test was used where indicated when two conditions were compared. The null hypothesis was rejected at a significance level of $P = 0.05$. For experiments with iPSC lines, biological replicates ($n$) are

defined as differentiations performed at least one cell-split apart, which is generally at least 1 week and each clone was differentiated three times.

**Ethical approval**. Ethical approval for the use of human tissue was obtained from the Oxford C REC Ethics Committee (No. 15/SC/0639).

**Reporting summary**. Further information on research design is available in the Nature Research Reporting Summary linked to this article.

## Data availability

RNASeq data were deposited in Gene Expression Omnibus (GEO) as follows: GSE149632 and GSE171999. Mass spectrometry data were deposited to the ProteomeXchange Consortium via the PRIDE partner repository with the dataset identifier PXD024198. Source data are provided with this paper.

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

## Acknowledgements

Brain tissue samples and associated clinical and neuropathological data were supplied by the Oxford Brain Bank, supported by the Medical Research Council (MRC), Brains for Dementia Research (BDR) and the NIHR Oxford Biomedical Research Centre and the Parkinson's UK Brain Bank, funded by Parkinson's UK, a charity registered in England and Wales (258197) and in Scotland (SC037554). We thank Dr. Sally Cowley, Head of the James Martin Stem Cell Facility, University of Oxford for providing iPSC cell stocks. We thank Raphael Heilig for assistance with mass spectrometry, which was performed at the Target Discovery Institute Mass Spectrometry facility. The study has received support from the Innovative Medicines Initiative 2 Joint Undertaking under grant agreement no. 116060 (IMPRiND). This Joint Undertaking receives support from the European Union's Horizon 2020 research and innovation programme and EFPIA. IMPRiND is supported by the Swiss State Secretariat for Education, Research and Innovation (SERI) under contract number 17.00038. GKT was funded by a Wellcome Trust Intermediate Clinical Fellowship (097479/Z/11/Z), the Wellcome Beit Prize (097479/Z/11/A), BMA Foundation Vera Down award, Alzheimer's Research UK and the National Institute for Health Research (NIHR) Oxford Biomedical Research Centre. R.M. was funded by EC Joint Programme on Neurodegenerative Diseases and Agence Nationale pour la Recherche (TransPathND, ANR-17-JPCD-0002-02 and Protest-70, ANR-17-JPCD-0005-01), the Grand Prix Scientifique Fondation Simone and Cino Del Duca and the JiePie 2019 award. This work has benefited from the Electron microscopy facility of Imagerie-Gif, supported by ANR-10-INBS-04-01 and the Labex ANR-11-IDEX-0003-02. The opinion expressed and arguments employed herein do not necessarily reflect the official views of these funding bodies.

## Author contributions

B.T., S.S.S., A.F., L.B., J.M. and C.Z. collected, analysed and interpreted the data. S.H.A. provided the isogenic iPSC line. D.A., R.F., D.S., R.M. and G.K.T. analysed and interpreted the data. B.T., S.S.S., R.M. and G.K.T. prepared the manuscript. B.T., R.M. and G.K.T. designed the research. G.K.T. was responsible for the overall design and coordination of the study.

## Competing interests

The authors declare no competing interests.
