## [Peer Review File · Nature Communications]

REVIEWER COMMENTS

Reviewer #1 (Remarks to the Author):

This manuscript addresses phenotypic changes in iPSC-derived dopaminergic neurons from two angles: mutations (SNCAA53T) and copy number variations (SNCATRIP) of the SNCA gene, as well as seeding with de novo fibrils and PMCA-amplified Parkinson's disease and MSA aggregates. The authors reported that by applying fibrils to the cell cultures, phosphorylated α -synuclein increased in a dose- and time-dependent manner, with the effects being most pronounced in SNCATRIP neurons compared to SNCAA53T or healthy controls. PD- and MSA-derived fibrils are also of different conformational strains, as exhibited by distinct proteinase K-resistant banding patterns. Upon applying de novo fibrils and brain-amplified aggregates to cell culture, the authors reported intracellular depositions of phosphorylated α -synuclein (and de novo, PD, or MSA seeds created distinct shapes), as well as nuclear fragmentation, again with SNCATRIP showing more pronounced results than healthy controls. Comparing baseline gene expression patterns between healthy controls and SNCATRIP neurons, the authors have found that major differences occur in oxidative phosphorylation pathway. SNCATRIP neurons were reported to have a reduction in basal oxygen consumption rate (OCR), maximal respiration, and ATP production, and these readouts were exacerbated after seeding with de novo fibrils or brain-amplified aggregates alike. Upon isogenic correction for the triplication, phosphorylated α -synuclein and fragmented nuclei count were significantly reduced, suggesting the role of gene expression in the observed phenotypes. Using BioID-mediated interactome analysis, the authors discovered that DJ-1, a deglycase, is a major interactor of α -synuclein. In particular, de novo fibrils interacted significantly with DJ-1, while PD and MSA-derived aggregates did not. The authors subsequently knocked out DJ-1 or its relative Glyoxalase-1 and found that both of these conditions, as well as saturating DJ-1 with methylglyoxal application, increased phosphorylated α -synuclein burden after seeding, suggesting a relationship between glycation status and protein aggregation as well as proposing a mechanism involving lysine-12 and ubiquitination.

While there is already literature comparing de novo fibrils and PD/MSA conformational strains, the authors have occupied a unique niche by looking at the three-way interaction between strains, genotypic predispositions, and cellular phenotypes. Of particular interest is that toxicity in seeding reactions increases with increasing SNCA expression as well as the first demonstration of potential differential interactomes between various synuclein strains. The authors' conclusions are, in general, adequately justified by their experimental data. Overall, this is a very interesting collection of important experiments. However, I do have some suggestions for improvement and clarity of presentation.

Major Points:

1. While the utility of using PMCA-amplified PD and MSA strains is obvious, it would be very beneficial to show that the properties of these amplified strains reflect those of the original brain-derived material. For instance, do the PK digestion patterns of the brain-derived aggregates match those of the amplified aggregates? What happens when non-amplified PD or MSA brain extract (non-diluted) is used to seed the SNCATRIP cells?
2. That SNCA mRNA levels increase in response to seeding is very interesting. The authors should discuss potential mechanisms by which this might occur.
3. Figure 1a needs to have control images added to it. In general, the exact definition of

“NSC” should be included in each figure legend. For example, in Figure 1b, is it untreated cells? Is it cells treated with monomeric synuclein?

4. Figure 4b is quite confusing. What are the log₂ ratios with respect to? Is it de novo fibrils vs. brain-derived (PD or MSA) fibrils? The same comment applies to Extended Data Table 3.

5. That DJ-1 preferentially becomes biotinylated in response to seeding using de novo fibrils is interesting. However, does this suggest that DJ-1 has no relevance for actual brain-derived human disease-associated synuclein strains? What happens when the DJ-1 knockdown cells are challenged with the PD- or MSA-amplified strains? This seems like a critical experiment to tie the data together.

Other minor issues:

In the first paragraph of the Introduction, the phrase “not life-limiting” when referring to PD is not a good wording choice. While PD is more slowly progressive than MSA and can be symptomatically managed more easily, it is still a disease that can have profound effects on a person’s well-being and shorten their lifespan.

In Figure 2c, it would be really useful for the reader if the authors included an image of a single gel that contains all of the samples at a single timepoint (say 60 min). This would allow for a much more direct comparison of banding patterns.

Figure 3b could use some higher magnification insets to better illustrate what the PSyn pathology looks like.

The legend for one of Figure 3e-g is missing.

In Figure 5b, the authors need to explain what the open circles vs. blue circles mean.

In Figure 5d and the associated text, the authors should clarify whether these cells are knocked-out or knocked-down for DJ-1 or GLO1. Extended Data 8 seems to suggest knock-down. Also, the legend for this figure needs to specify that de novo fibrils were used as the seeding species.

Extended data figure 2: Would the authors please comment why the oligo α -synuclein graph is at 1.5x? What are these 1.5-fold in comparison to?

Fig 1d (and all related figures): The in-text discussion uses “weeks after seeding” but the plots uses DIV as units. Inferring 14 days as 2 weeks is easy but DIV 59 as 2 weeks after seeding requires some work for the reader. The authors may wish to consider relabeling all the plots to “weeks after seeding,” and just discuss the specific DIV for each week in the materials and methods.

page 6, 8 lines from the bottom: Please elaborate on the optimization experiments used to decide on the dilutions used and time points for second/third cycles, either here or in the materials and methods. It would be helpful if authors could provide some of the data collected from the optimization process as well to support their experimental decisions.

Extended Data figure 5: looks like the figure for a is missing... The legend describes this missing figure as a comparison between HC and SNCATRIP pSYN levels (without fold-

change normalization) under the various seeding conditions? Unless the authors wish to refer to main figure 4c – in which case ED figure 5 legend part a should be removed, and b&c needs to be changed to a&b.

Page 13, 2nd paragraph: Is this the first time “isogenic correction” is mentioned in the paper? If so, the authors may wish to briefly tell the readers how this correction was done as well as where the cell lines came from, in addition to the paper citation they have already made.

Extended Figure 7b: It would be appreciated if R and K in the workflow are defined in the figure legend.

That a Venus-tagged synuclein construct was used for the experiments involving DJ-1 and GLO1 knockdown needs to be mentioned in the main text.

Reviewer #2 (Remarks to the Author):

This manuscript reports aggregated α -synuclein in iPSC-derived dopaminergic neurons following treatment with exogenous α -synuclein pre-formed fibrils (PFFs). The authors observed a time-dependent increase in aggregated α -synuclein, which was greatest in lines exhibiting SNCA triplication, followed by lines expressing A53T α -synuclein, and finally wt lines. In addition, the authors observed that α -synuclein expression was increased.

Exposure to different PFF conformers (termed fibrils and ribbons) resulted in different levels of aggregated α -synuclein. Furthermore, PMCA using PD or MSA brain homogenates produced different PFF conformers (which were also distinct from fibrils and ribbons). Similar amounts of aggregated α -synuclein in iPSC-derived dopaminergic neurons were detected following treatment with exogenous PD and MSA PMCA PFFs, but the morphology of the aggregates differed. Unlike fibrils, PD and MSA PMCA PFFs were neurotoxic in SNCA triplication lines (but not in wt lines), with MSA PMCA PFFs the most neurotoxic. Isogenic correction of SNCA triplication led to a reduction in aggregated α -synuclein and neurotoxicity.

Transcriptomic profiles differed between SNCA triplication and wt lines and were generally related to SNCA expression and mitochondrial respiration. Deficits in mitochondrial respiration were observed in SNCA triplication lines compared to wt lines, which was increased following exposure to PFFs.

Proximity-labelling revealed different interactomes of aggregated α -synuclein in HEK cells following treatment with different exogenous PFF conformers. An increased interaction with DJ-1 was observed in HEK cells treated with fibrils compared to PMCA PFFs. DJ-1 and Glo-1 KO HEK cell lines exhibited increased glycation of α -synuclein more aggregated α -synuclein upon exposure to fibrils. The authors hypothesise that glycation blocks ubiquitination and thereby prevents turnover of α -synuclein. Methylglyoxal treatment of iPSC-derived dopaminergic neurons increased the amount of α -synuclein aggregates and neurotoxicity upon exposure to PFFs.

The data is of high quality, the statistical analyses are appropriate and, in general, there is sufficient detail to enable a researcher to reproduce the work.

From these experiments, the authors conclude that 1.) disease-relevant/specific conformations/polymorphs were templated/imprinted onto endogenous α -synuclein in both the PMCA and PSC-derived dopaminergic neurons exposed to PMCA PFFs, 2.) seeded aggregation and toxicity of α -synuclein PFFs depends on the concentration of the soluble protein and PFF conformation, 3.) seeded aggregation of α -synuclein fibrils increases expression of the protein, and d.) the interactome of seeded aggregates depends on fibril conformation.

The first three conclusions support previously published studies (see for example references 3-7, 15 and 26-31 in the manuscript), extending them to human cell lines. The final conclusion that the interactome of seeded aggregates depends on fibril conformation is novel, of interest to others in the field and is likely to influence thinking in the field. Emerging evidence suggests that the conformation of aggregated α -synuclein may contribute to the pattern of neurodegeneration and subsequent clinical phenotype of synucleinopathies. However, the underlying mechanisms are not known. This work provides evidence that distinct interactomes between different aggregate conformers may play a role.

I only have the following additional comments,

1. From the results presented, it is not possible to conclude that disease-relevant/specific conformations/polymorphs were templated/imprinted onto endogenous α -synuclein. This would require directly comparing the conformation of the seeds and the seeded aggregates (i.e. the aggregates within the PD and MSA brain homogenates compared to the aggregates generated by PMCA; and the PFFs compared to the aggregates in the iPSC-derived dopaminergic neurons/ HEK cells). Unless the authors can provide such evidence, such as those analyses used in Fig 3b and c, I recommend removing any reference to templating/imprinting of conformations/polymorphs.

2. From the results presented, it appears that the aggregated α -synuclein in iPSC-derived dopaminergic neurons following treatment with PFFs could arise either from the uptake and phosphorylation of the PFFs themselves or from the aggregation and phosphorylation of endogenous α -synuclein. Can the authors definitively distinguish between these two possibilities? If not, I recommend that a sentence acknowledging this is added.

3. The legend for Fig 1 should state which methods were used to generate the data (i.e. immunohistochemistry or Western blot or FRET)

4. The Western blot in Fig 1b and e is cropped such that only 15 kDa α -synuclein is shown. Were there any higher molecular mass bands, such as those representing ubiquitinated α -synuclein? These should also be shown if present.

Reviewer #3 (Remarks to the Author):

In this work Tanudjojo and colleagues have generated synuclein fibril assemblies from PD and MSA patient brains and used these assemblies to treat iPSC derived dopaminergic neural cultures and HEK293 cells. In a series of fairly straightforward experiments, this work confirms much of what has been presented from the growing set of studies adding exogenous aggregated synuclein to various cell types. The interaction of endogenous

abnormal synuclein with exogenous pathological forms is important for the field to know and this represents a nice in vitro model to explore mechanisms. There is a slight lack of novelty given the range of studies already done in this area and some overlap with existing work. however, there does seem to be enough new material to make this an important study.

There are a few points that would help add clarity and context to the work.

1. First and most importantly, it is critical to describe what is known about the origin of disease for the patients the fibril assemblies were derived from. Are these genetic cases that harbor A53T or other synuclein mutations? This information would drastically alter the interpretation of the work.
2. Given the size and toxicity differences of the PD and MSA derived fibrils, is the increased toxicity of the shorter MSA fibrils based on fibril length (as has been previously suggested) or some other structural difference in the assembly? A shorter control fibril would help to address this.
3. Please present p-Syn westerns where this data is presented (as in fig 1). Example western bands are missing from figure 3a, figure 4f ect.
4. The treatment of all p-129 synuclein as an aggregate is potentially confounding. The authors should confirm synuclein aggregation through additional means.
5. The RNA-Seq presented in figure 4 would be more informative if it was analyzed to highlight differences (or lack thereof) incurred on fibril addition. The differential expression analysis of the triplication lines and controls is not very informative in the context of the work. Pointing out non-significant terms in the GSEA is neither helpful or informative as no conclusions can be drawn. It is interesting that no substantial gene expression changes were incurred on fibril addition. Was SNCA expression altered in this analysis as presented in figure 1F? Are there time dependent changes immediately after fibril addition that are missed by checking only two weeks post addition?

Minor points:

1. In figure 3 what is the difference in fibrils from PD patient 3C and 3G? this does not seem to be specified in the text.
2. In Figure 3 e-g the y-axis are labeled as fold change yet given as a percent (interpreted as a % of the untreated cells). Do the authors mean to suggest that they observe 25 fold loss of viable nuclei in e or a 25% loss?
3. Extended data table 3 is appended unlabeled to the end of the article file instead of in the extended data file.

RESPONSE TO REVIEWERS

We would like to thank the reviewers for their thoughtful comments, which have enabled us to strengthen our manuscript, and the Editor for giving us the opportunity and time to address their questions during this unprecedented period of restricted lab work as a result of the pandemic. We enclose our detailed responses below supplemented by extensive new data (including the generation of iPSC DJ-1 KO clones and further transcriptomic and biochemical analyses) or changes in the text as indicated. In response to specific comments:

Reviewer #1:

While there is already literature comparing de novo fibrils and PD/MSA conformational strains, the authors have occupied a unique niche by looking at the three-way interaction between strains, genotypic predispositions, and cellular phenotypes. Of particular interest is that toxicity in seeding reactions increases with increasing SNCA expression as well as the first demonstration of potential differential interactomes between various synuclein strains. The authors' conclusions are, in general, adequately justified by their experimental data. Overall, this is a very interesting collection of important experiments.

We thank the reviewer for these comments

1. While the utility of using PMCA-amplified PD and MSA strains is obvious, it would be very beneficial to show that the properties of these amplified strains reflect those of the original brain-derived material. For instance, do the PK digestion patterns of the brain-derived aggregates match those of the amplified aggregates? What happens when non-amplified PD or MSA brain extract (non-diluted) is used to seed the SNCATRIP cells?

The reviewer raised a point that we carefully considered. The main purpose of our study is the development of a “reductionist” model that bypasses the limitation of using brain extract as this consists of aggregated alpha-synuclein *as well as* a number of other proteins, lipids and potentially cellular debris that non-specifically associate with fibrillar assemblies even after sarkosyl extraction. There is also considerable variability between brains in terms of these associations. It would therefore be impossible to accurately interpret whether any cellular or molecular phenotype we observed is caused by brain isolated strains or associated contaminants. The use of undiluted extract would be even more complicated to interpret in the context of our study. In this respect, experiments with brain extract would not add clarity to our conclusions and are beyond the scope of the work presented in this study with brain amplified fibrils.

In our experience, Proteinase K digestion directly from human brain cannot be compared vis a vis to the digestion pattern of an amplified recombinant fibril because the post-mortem delay and post-mortem activation of proteolytic enzymes or effects of other brain constituents on Proteinase K activity would mean that it is not possible to have highly controlled conditions for the comparison between the digestion patterns of brain extra vs amplified fibril. Indeed, though PMCA is a widely accepted methodology for amplification and study of proteopathic seeds previous publications (e.g. Shahnawaz et al., Nature 2020; Strohaker et al., Nature Commun 2019; Saborio et al., Nature 2001) did not provide a direct comparison between the seed isolated from brain or CSF (without amplification) and the resulting amplified product by Proteinase K digestion. Beyond the caveats mentioned above, such experiments require a considerable amount of human tissue for extraction of fibrils or preparation of extracts that is not readily available.

To provide further experimental evidence for our conclusions, we have now performed Proteinase K digestion using the lysate from iPSC-derived neurons 2 weeks after seeding. This experiment showed that aggregated α -synuclein in PD-seeded neurons was more resistant to Proteinase K digestion than MSA-seeded neurons based on immunoblotting with a mixture of three anti- α -synuclein antibodies (ASynM, 4B12, 10D2) after limited proteolysis (included in Figure 3 panel e)

as we detected with Proteinase K digestion of PD- vs MSA-amplified fibrils stained with Coomassie blue (shown in Fig. 2c and Extended Data Fig. 6). This finding suggests that the conformation of iPSC neuronal aggregates reflects the pattern of those amplified from human brain.

We have also modified the title (by including the word “amplified”) and text to more accurately reflect the scope of our study and focus of our conclusions which would not be altered by the experiments with brain extract: “Phenotypic manifestation of α -synuclein strains amplified from Parkinson’s disease and multiple system atrophy in human dopaminergic neurons”. In addition, as suggested by the reviewer 2 and explained further below, we have replaced reference to “disease-relevant conformations/polymorphs” with “polymorphs or strains amplified from PD or MSA” as shown by tracked changes.

2. That SNCA mRNA levels increase in response to seeding is very interesting. The authors should discuss potential mechanisms by which this might occur.

We agree with the reviewer that this is an interesting finding. One potential explanation is epigenetic dysregulation. A number of transcription factors have been implicated in SNCA regulation including ZSCAN21 and GATA2 (Brenner et al., 2015; Scherzer et al., 2008) and DNA hypomethylation has been detected in the substantia nigra of PD and DLB brains (Jowaed et al., 2010; Matsumoto et al., 2010). SNCA expression is dependent on intron 1 methylation, and inhibition of methylation in this region would result in increased expression. α -Synuclein aggregation was shown to sequester DNA methyltransferase 1 (DNMT1) away from the nucleus in PD brains and mouse models alike (Desplats et al., 2011), providing one mechanism that could increase SNCA mRNA. Alternatively, the presence of pathogenic α -synuclein aggregates in the cytosol may interfere with miRNA processing. For example, microRNAs 34b and 34c, that were shown to repress SNCA expression (Villar-Menéndez et al., 2014) were found to be down-regulated in the substantia nigra (SN) and other regions in PD patients with Lewy pathology (Miñones-Moyano et al., 2011). We briefly mentioned these mechanisms in page 12, line 7.

3. Figure 1a needs to have control images added to it. In general, the exact definition of “NSC” should be included in each figure legend. For example, in Figure 1b, is it untreated cells? Is it cells treated with monomeric synuclein?

We have now modified this figure to include control images as requested by the reviewer and clarified in all the legends that NSC stands for non-seeded control which means untreated cells. In fact, we tested and confirmed that monomeric alpha-synuclein is comparable to untreated cells in terms of aggregate detection and these data are now included in Fig. 3, panel b. We also showed that monomer, similarly to untreated cells, is not toxic to neurons as shown in Fig. 3 panel i.

4. Figure 4b is quite confusing. What are the log2 ratios with respect to? Is it de novo fibrils vs. brain-derived (PD or MSA) fibrils? The same comment applies to Extended Data Table 3.

We assume the reviewer is referring to Figure 5b and Extended Data Table 3. We used peptide precursor ion abundance to compare the volume under the chromatographic signal of each peptide detected for a protein. Peptide signals were then summarised into ratios for each protein across the different conditions, yielding a fold change value. Specifically, the ratio of MSA or PD to *de novo* fibrils was first calculated and then converted to log2. These data were analysed with Maxquant followed by statistical analysis within the biological replicates to yield fold changes and multiple testing corrected p-values. The plot is demonstrating the reproducibility of the comparison. It is expected that most data points are on the diagonal as the log ratio in both replicates should be identical for a robust measurement. α -Syn is located at the origin of the graph which means that it is not regulated in both of the replicates, while DJ-1 is up-regulated by a factor of 4 on both

replicates. Proteins outside the diagonal were only found to be regulated in one of the replicates and should be regarded with caution. We have clarified this in the methods section as shown in page 20, lines 6-11. We have also deposited the mass spectrometry proteomics data to the ProteomeXchange Consortium via the PRIDE partner repository with the dataset identifier PXD024198.

5. That DJ-1 preferentially becomes biotinylated in response to seeding using de novo fibrils is interesting. However, does this suggest that DJ-1 has no relevance for actual brain-derived human disease-associated synuclein strains? What happens when the DJ-1 knockdown cells are challenged with the PD- or MSA-amplified strains? This seems like a critical experiment to tie the data together.

We would like to thank the referee for raising this pertinent question. We agree that this is a critical experiment that we have now addressed in two complementary experiments:

Firstly, we have confirmed in our clonal HEK line that CRISPR/Cas9 mediated knockout of DJ-1 or GLO-1 (which functions in the same deglycation pathway as shown in Fig 5d) increased seeded aggregation when the cells were challenged with PD- or MSA- amplified strains in addition to *de novo* fibrils (Figure 5, panels e, f for DJ-1 and panels h, i for GLO-1). These experiments involved the quantification by immunoblotting of pSyn in RIPA insoluble fractions following cell lysis and ultracentrifugation. This analysis showed that aggregation is increased in DJ-1 or GLO-1 knockout cells compared to ones transduced with non-target sgRNAs (two sgRNAs per condition). We also now show that knockout of DJ-1 or GLO-1 was also associated with aggregate-induced toxicity based on LDH measurements in corresponding conditioned media (Figure 5 panels g and j). These data reinforce the notion that DJ-1 may act in the same deglycation pathway as GLO-1 and complement our finding that Methylglyoxal treatment in iPSC-derived neurons also increased aggregation and toxicity (Fig 5, panels m and n).

Secondly, to fully validate the relevance of our observation in human neurons, we have now generated CRISPR/Cas9 mediated DJ-1 knockout clones in our SNCA Triplication line (Figure 6 and Extended Data Figures 13-15). Three clones were selected (2 resulting from exon 2 disruption and 1 resulting from exon 4 disruption) and examined in three independent differentiations. These experiments showed that when DJ-1 is knocked out, aggregates induced with all three strains are now toxic and accelerate cell loss and LDH release in the conditioned media (new Figure 6). Given that loss of function mutations in DJ-1 cause PD with Lewy body (i.e. α -synuclein-immunoreactive) pathology (Taipa et al., Brain 2016), our data provide support for the relevance of DJ-1 in α -synuclein homeostasis and aggregation in human neurons.

6. In the first paragraph of the Introduction, the phrase “not life-limiting” when referring to PD is not a good wording choice. While PD is more slowly progressive than MSA and can be symptomatically managed more easily, it is still a disease that can have profound effects on a person’s well-being and shorten their lifespan.

We agree with the reviewer and we have now removed the words “not life-limiting” from the sentence.

7. In Figure 2c, it would be really useful for the reader if the authors included an image of a single gel that contains all of the samples at a single timepoint (say 60 min). This would allow for a much more direct comparison of banding patterns.

This has now been added as requested by the reviewer for two timepoints (15min and 60min) as Extended Data Figure 6.

8. Figure 3b could use some higher magnification insets to better illustrate what the pSyn pathology looks like.

This has now been included in the revised Figure 3 that is now labelled as Panel c

9. The legend for one of Figure 3e-g is missing.

We thank the reviewer for pointing out this omission which we have now corrected the legend for the revised Figure 3.

10. In Figure 5b, the authors need to explain what the open circles vs. blue circles mean.

The open circles represent all the proteins (1002 in total) identified across conditions whereas the blue circles represent the differentially regulated proteins as also explain in our response to point 4 above. This information has now been added in the legend.

11. In Figure 5d and the associated text, the authors should clarify whether these cells are knocked-out or knocked-down for DJ-1 or GLO1. Extended Data 8 seems to suggest knock-down. Also, the legend for this figure needs to specify that de novo fibrils were used as the seeding species.

We have now used anti-DJ-1 antibodies that are commercially available to assess the lysates from CRISPR/Cas9 mediated DJ-1 knockout cells for previous and new experiments. Two sgRNA were used per condition (non-target, DJ-1 or GLO1). These experiments were repeated to compare *de novo* Fibrils to PD or MSA amplified strains. This analysis is included in Fig. 5 panel e and shows that DJ-1 is successfully knocked out. Similarly, we have also performed immunoblotting to assess levels of GLO-1 which also confirmed successful knockout (Fig. 5 panel h). We have now also generated and characterised DJ-1 knockout iPSC clones as shown in Fig. 6.

12. Extended data figure 2: Would the authors please comment why the oligo α -synuclein graph is at 1.5x? What are these 1.5-fold in comparison to?

The values in the previous Extended data Figure 2 panel a were erroneously represented as fold change to negative control and we thank the reviewer for pointing this out. The negative control in this assay consisted of cell lysis buffer with donor and receptors antibodies without cell lysate. This panel has now been corrected to represent fold change to healthy control (HC). This is now labelled as Extended Data Figure 4.

13. Fig 1d (and all related figures): The in-text discussion uses “weeks after seeding” but the plots uses DIV as units. Inferring 14 days as 2 weeks is easy but DIV 59 as 2 weeks after seeding requires some work for the reader. The authors may wish to consider relabeling all the plots to “weeks after seeding,” and just discuss the specific DIV for each week in the materials and methods.

We agree with the reviewer’s helpful suggestion and we have relabelled all the figures as recommended

14. Page 6, 8 lines from the bottom: Please elaborate on the optimization experiments used to decide on the dilutions used and time points for second/third cycles, either here or in the materials and methods. It would be helpful if authors could provide some of the data collected from the optimization process as well to support their experimental decisions.

This information is included in Figure 2 but we have now added the following clarification in the Methods section (page 24, lines 24-27): “The time at which an aliquot from one amplification reaction was withdrawn for a subsequent reaction was defined as the time where ThT fluorescence increases for amplification reactions performed with PD and MSA patients brain homogenates,

and not in reactions containing control patients brain homogenates or monomeric α -synuclein alone”.

15. Extended Data figure 5: looks like the figure for a is missing... The legend describes this missing figure as a comparison between HC and SNCA^{TRIP} pSYN levels (without fold-change normalization) under the various seeding conditions? Unless the authors wish to refer to main figure 4c – in which case ED figure 5 legend part a should be removed, and b&c needs to be changed to a&b.

We thank the reviewer for pointing this out and we have now corrected the caption. This figure is now Extended data Figure 9 in the revised Supplement.

16. Page 13, 2nd paragraph: Is this the first time “isogenic correction” is mentioned in the paper? If so, the authors may wish to briefly tell the readers how this correction was done as well as where the cell lines came from, in addition to the paper citation they have already made.

We have now included a paragraph under Methods, “CRISRP/Cas9 modification of iPSC lines” section as shown in pages 17-18, to briefly summarise the correction which is described in detail in the relevant cited publication (Heman-Ackah et al., 2017, Hum. Mol. Gen. 22:4441-4450). In brief, Cas9 nickase sgRNAs were designed to target *SNCA* exon 4 and applied to the ND34391G iPSC line (obtained from the Coriell Institute, distributed through the NINDS Fibroblasts and iPSCs Collection). After clonal expansion of Cas9 nickase-transfected iPSCs, four clones were identified as variants by high resolution melt analysis and one clone was confirmed by sequencing as a double knockout. The double knockout clone (i.e. deletion of the two additional *SNCA* gene copies) was sequenced for off-target effects and quality controlled¹¹. α -Synuclein mRNA and protein levels in the isogenic iPSC line were tested and shown to be reduced to levels comparable to the healthy control iPSC line¹¹. In this context, isogenic refers to the fact that the parental *SNCA* triplication iPSC line and the double knockout clone possess identical genetic background, which includes overexpression of the other genes in the triplication locus, and selective reduction of *SNCA* gene dosage to achieve normal expression levels.

17. Extended Figure 7b: It would be appreciated if R and K in the workflow are defined in the figure legend.

We thank the reviewer for this suggestion. We have now added this information as follows: R=Arginine and K=Lysine. This is now labelled Extended Data Figure 10 panel b

18. That a Venus-tagged synuclein construct was used for the experiments involving DJ-1 and GLO1 knockdown needs to be mentioned in the main text.

This information was included in the Methods and now also added in the main text as shown in page 10 line 17-18. In addition, we have now knocked out DJ-1 in the *SNCA* Triplication iPSC line to further validate the relevance of DJ-1 in human neurons as shown in Figure 6.

Reviewer #2

The data is of high quality, the statistical analyses are appropriate and, in general, there is sufficient detail to enable a researcher to reproduce the work.

We are grateful to the reviewer for their comment.

1. From the results presented, it is not possible to conclude that disease-relevant/specific conformations/ polymorphs were templated/imprinted onto endogenous α -synuclein. This

would require directly comparing the conformation of the seeds and the seeded aggregates (i.e. the aggregates within the PD and MSA brain homogenates compared to the aggregates generated by PMCA; and the PFFs compared to the aggregates in the iPSC-derived dopaminergic neurons/ HEK cells). Unless the authors can provide such evidence, such as those analyses used in Fig 3b and c, I recommend removing any reference to templating/imprinting of conformations/polymorphs.

As stated in our response to point 1 by reviewer #1 above, in our experience, Proteinase K digestion of human brain extract is not directly comparable to the digestion pattern of an amplified recombinant fibril because post-mortem delay and post-mortem activation of proteolytic enzymes or effects of other brain components on Proteinase K activity would mean that it is not possible to have highly controlled conditions for direct comparison between digestion patterns of brain extra vs amplified fibril. These experiments would also require a considerable amount of human tissue for extraction of fibrils that unfortunately is not readily available. Although PMCA is an established methodology for amplification of brain seeds (e.g. Nature 2002, 2019, Nature Comm 2019), none of these publications provided a direct comparison between endogenous brain-derived seeds and amplified ones with Proteinase K digestion. As suggested by the reviewer, we have removed reference to “disease-relevant conformations/polymorphs” which are instead referred to as “polymorphs or strains amplified from PD or MSA”. We agree with the reviewer that such change in the text more accurately reflects the scope of our study and we thank them for their suggestion.

We have successfully performed Proteinase K digestion of neuronal lysates 2 weeks after seeding of iPSC-derived neurons as lysates were snap frozen and processed under controlled conditions. The lysate was solubilised and subjected to Proteinase K digestion followed by immunoblotting with a mixture of the following anti-synuclein antibodies: ASyM (clone number 4.2, Agrisera, AS13 2719, residues 1-15), 4B12 (Biolegend, cat # 807801, residues 103-108) and 10D2 (Merck cat # MABN633, residues 118-127). These antibodies were chosen because they cover well the α -Syn primary structure. This experiment showed that aggregated α -synuclein in PD-seeded neurons was more resistant to Proteinase K digestion than MSA-seeded neurons based on immunoblotting with the mixture of the three anti- α -synuclein antibodies after limited proteolysis (included in Figure 3 panel e) as we detected with Proteinase K digestion of PD- vs MSA-amplified fibrils stained with Coomassie blue (shown in Fig. 2c and Extended Data Fig. 6). This finding suggests that the conformation of iPSC neuronal aggregates reflects the pattern of those amplified from human brain.

2. From the results presented, it appears that the aggregated α -synuclein in iPSC-derived dopaminergic neurons following treatment with PFFs could arise either from the uptake and phosphorylation of the PFFs themselves or from the aggregation and phosphorylation of endogenous α -synuclein. Can the authors definitively distinguish between these two possibilities? If not, I recommend that a sentence acknowledging this is added.

We agree with the reviewer that clarification of this issue will strengthen the conclusions of our study. We have now addressed this issue by using polymorph “fibrils” with Ser129 mutated to alanine (S129A) that cannot be phosphorylated. We showed in both iPSC-derived neurons and HEK cells that the signal intensity of pSer129 generated by S129A seeds is similar to fibrils generated from wildtype α -synuclein seeds. These data are included as Extended data Figure 3 and suggest that irrespective of the cell type, the signal arises from endogenously aggregated α -synuclein. This has now been added to the main text page 4 line 22-26.

3. The legend for Fig 1 should state which methods were used to generate the data (i.e. immuno-histochemistry or Western blot or FRET).

We have now explicitly stated in all legends which data were generated by Homogeneous Time Resolved Fluorescence (HTRF), immunoblotting or immunostaining.

4. The Western blot in Fig 1b and e is cropped such that only 15 kDa α -synuclein is shown. Were there any higher molecular mass bands, such as those representing ubiquitinated α -synuclein? These should also be shown if present.

We have now included the full blot that indeed included a higher molecular weight band of α -synuclein in Fig 1 now labelled panel c. Based on the size (~28kDa) it is not possible to conclude whether this reflects di-ubiquitinated α -synuclein (14kDa + 7 kDa for each ubiquitin) or dimeric form of the protein following SDS denaturation and boiling of the aggregates prior to immunoblotting.

Reviewer #3:

1. In this work Tanudjojo and colleagues have generated synuclein fibril assemblies from PD and MSA patient brains and used these assemblies to treat iPSC derived dopaminergic neural cultures and HEK293 cells. In a series of fairly straightforward experiments, this work confirms much of what has been presented from the growing set of studies adding exogenous aggregated synuclein to various cell types. The interaction of endogenous abnormal synuclein with exogenous pathological forms is important for the field to know and this represents a nice in vitro model to explore mechanisms. There is a slight lack of novelty given the range of studies already done in this area and some overlap with existing work. however, there does seem to be enough new material to make this an important study.

We thank the reviewer for recognising the importance of our study.

2. First and most importantly, it is critical to describe what is known about the origin of disease for the patients the fibril assemblies were derived from. Are these genetic cases that harbor A53T or other synuclein mutations? This information would drastically alter the interpretation of the work.

The fibrils were derived from patients with sporadic PD and did not harbour any known α -synuclein mutations. These cases were sequenced as part of a previous genetic study (Guella et al., Ann Neurol 2016; 79:991-999). We have now amended the Table to state under "Diagnosis" that PD cases were sporadic and the corresponding legend as follows: "Each case was extensively characterised neuropathologically including staining for other proteinopathies and was not associated with *SNCA* mutations or multiplications."

2. Given the size and toxicity differences of the PD and MSA derived fibrils, is the increased toxicity of the shorter MSA fibrils based on fibril length (as has been previously suggested) or some other structural difference in the assembly? A shorter control fibril would help to address this.

The reviewer raised a point that we carefully considered. As extensively documented in the field, fibrils generated *de novo* or amplified from brain are sonicated in order to facilitate entry into cells. Our preparations were sonicated under strictly controlled conditions. The sonicated fibrils were checked for size by TEM and found to have similar (60nm on average) length in all strains used. This is now added as Extended Data Figure 8 showing the fibrils and quantification of their length after sonication. Therefore, the shorter aggregates generated inside the neurons are unlikely to arise from differences in the length of the fibrils used as seeds.

3. Please present p-Syn westerns where this data is presented (as in fig 1). Example western bands are missing from figure 3a, figure 4f ect.

We wish to point out that data in Fig. 1d, 1e, 1h, Fig. 3a, 3b, Fig. 4f, Fig. 5m and Fig. 6k were generated using HTRF analysis and do not represent average of Western blot analysis. We have now clarified this in the corresponding legend to avoid any misinterpretation by readers. We confirm that the revised manuscript contains representative examples for all experiments where immunoblotting was performed (Fig. 1c, Fig. 1f, Fig 3e, Fig. 4e, Fig. 5c, Fig. 5e, Fig. 5h, Fig. 5k, Fig. 6b, Fig. 6e).

4. The treatment of all p-129 synuclein as an aggregate is potentially confounding. The authors should confirm synuclein aggregation through additional means.

We agree with the reviewer that clarifying this issue is relevant to the interpretation of our results. We have now fractionated iPSC-derived neuron lysates into RIPA-soluble and RIPA-insoluble fraction following 100,000g centrifugation. These fractions were analysed by immunoblotting relative to total protein loading as shown using Ponceau and quantified. This analysis showed that p-S129 synuclein (pSYN) signal is almost exclusively (97%) in the insoluble fraction detected as monomer and smear, indicating that this readout is a corollary of aggregated alpha-synuclein (Fig. 1c and Extended Data Fig. 2). We have also shown by immunofluorescence and confocal microscopy that pSYN positive inclusions co-localise with p62 (Fig. 1, panel b), which is another widely used marker that is sequestered in aggregates in human neurodegenerative diseases, including Lewy bodies.

5. The RNA-Seq presented in figure 4 would be more informative if it was analyzed to highlight differences (or lack thereof) incurred on fibril addition. The differential expression analysis of the triplication lines and controls is not very informative in the context of the work. Pointing out non-significant terms in the GSEA is neither helpful or informative as no conclusions can be drawn. It is interesting that no substantial gene expression changes were incurred on fibril addition. Are there time dependent changes immediately after fibril addition that are missed by checking only two weeks post addition?

We agree with the point made by the reviewer regarding the comparison of Triplication lines and Healthy controls at baseline and we have removed this dataset. We have now repeated RNASeq at two timepoints (early and late) as requested by the Reviewer using 1 μ M seeds (Fig. 4 a, b, c) which induces higher aggregate load compared to 0.1 μ M used previously (now included as Extended Data Fig. 10 Panel c). We specifically chose week 1 post seeding as this is the earliest timepoint we observed consistent signal by HTRF reflecting endogenous α -synuclein aggregation (Fig. 1e) and we repeated the week 2 timepoint because it is a timepoint where aggregation is increased further (Fig. 1e) but aggregate-induced neuronal loss is minimal and similar across conditions (Fig. 3g). We believe that timepoints earlier than 7 days could be compounded by changes in response to the acute addition of exogenous fibrils rather than the cellular response to intraneuronal aggregation. To minimise the variability between clones we have now tested one SNCA^{TRIP} clone three times at baseline or after treatment with the three different polymorphs (Fibrils, PD-amplified and MSA-amplified strains, total n=24 samples). Principle component analysis (PCA) on the 1000 most variable protein-coding genes showed that samples clustered by time (week 1 vs week 2) and treatment (aggregation versus baseline) but not by strain used to induce aggregation (Figure 4, panel a). Although a small number of differentially expressed genes were detected (Figure 4, panel b), there was no significant difference in any specific pathway in response to aggregation, in agreement with our earlier conclusion (Extended data Figure 10 panel c). To identify potential cellular responses to α -synuclein aggregation, we performed Gene Set Enrichment Analysis (GSEA) using Gene Ontology (GO). Although no pathway reached the 5% significance threshold following multiple testing correction, it is noteworthy that biologically relevant pathways were differentially up-regulated at week 1 and week 2 post seeding. Specifically, among the top 20 GO terms, Unfolded Protein Response and Endoplasmic Reticulum stress, endosomal

and endosome to Golgi transport were up-regulated at week 1 whereas protein folding, membrane protein proteolysis, mitochondrial membrane protein insertion, organelle fusion and apoptosis were detected in week 2. These pathways that have been implicated in PD pathogenesis, suggest a temporal (i.e. time-dependent) sequence of intracellular defects around organelle membrane function that may underpin the progressive neuronal dysfunction and eventually death caused by α -synuclein aggregation. We have now included this information in the revised Figure 4 and page 8.

6. Was SNCA expression altered in this analysis as presented in figure 1F?

We documented up-regulation of SNCA expression at 2 weeks post-seeding with 1 μ M fibrils (Fig 1g) by qPCR whereas in the initial RNASeq analysis we induced aggregation with 0.1 μ M of fibrils to assess the global response to subtoxic levels of aggregation. Therefore, the two conditions were not directly comparable. We have now examined the latest RNASeq data at 2 weeks post seeding with 1 μ M fibrils for SNCA mRNA. At this timepoint, we observed a trend for higher SNCA levels after aggregation but this did not reach statistical significance due to count variability between the two conditions. It should be noted that our second RNASeq experiment (n=3 samples per condition, for each of the 4 treatment conditions at each timepoint, total n=24 samples) was designed to address global transcriptomic changes at early versus late time points in response to point No 5 above raised by the reviewer. Therefore, when looking for one specific gene versus genome-wide correction at one timepoint this specific experiment may not have the same power as the qPCR analysis relative to the number of biological replicates available (n=9 samples per condition in qPCR with some variability as seen in the Fig. 1g).

Minor points:

1. In figure 3 what is the difference in fibrils from PD patient 3C and 3G? this does not seem to be specified in the text.

For this patient we included an additional brain region (Temporal Gyrus-G) that was examined at the initial characterisation of the amplified seeds. All analyses in iPSC were performed from PD fibrils derived from the same region (Cingulate). This was mentioned in Extended Data Figure 5 and now clarified in the main text in page 5, lines 28-29.

2. In Figure 3 e-g the y-axis are labelled as fold change yet given as a percent (interpreted as a % of the untreated cells). Do the authors mean to suggest that they observe 25fold loss of viable nuclei in e or a 25% loss?

In the new Figure 3, panels g, h, i are now labelled as “% viable nuclei” on the y-axis. These data were analysed as fold change to their corresponding non-seeded control group and are presented in the graph as percentage drop. For example, the non-seeded control group is defined at 100%, therefore values at the 75% mark signify a 25% loss.

3. Extended data table 3 is appended unlabelled to the end of the article file instead of in the extended data file

This excel file was uploaded separately, which may have caused the issue raised by the reviewer. We have added more information in the description of this file and also deposited the mass spectrometry proteomics data to the ProteomeXchange Consortium via the PRIDE partner repository with the dataset identifier PXD024198.

REVIEWER COMMENTS

Reviewer #1 (Remarks to the Author):

The authors have done an admiral job at addressing my comments, and I believe that the quality of the manuscript has improved significantly.

I have one additional comment for the authors: I believe that it was wise to change the wording in the paper to reflect that the experiments involve "strains amplified from MSA or PD patients". However, I think the authors need to briefly mention (probably in the Discussion) the recent findings from Sjors Scheres group on the structures of brain-derived vs. amplified MSA alpha-synuclein aggregates ("Seeded assembly in vitro does not replicate the structures of α -synuclein filaments from multiple system atrophy", FEBS Open Bio, 2021).

Reviewer #2 (Remarks to the Author):

It is my view that the authors have adequately addressed the points made by the three reviewers, with one exception. More detail is provided below.

The authors have acknowledged that it is not possible to conclude that disease-relevant/specific conformations/ polymorphs were templated/imprinted onto endogenous α -synuclein in their PMCA experiments and have thus removed such references. These references have been replaced with the terms "polymorphs/ strains amplified from PD/ MSA". To me, the term 'amplified from' indicates that more of something already in existence has been produced (in this case PD/ MSA strains). It would be better to use a phrase that does not imply production of the MSA/PD strain, for example, 'amplified/ produced in the presence of PD/ MSA.'

The experiments using PK digestion of neuronal lysates are a good addition to the manuscript and support the hypothesis that templated seeded aggregation is occurring in their cell model.

The experiments using S129A synuclein expressing cells are also a very welcome addition to the manuscript and support the hypothesis that the pS129 signal arises from cellular synuclein.

The legend for Figure 1 now states the methods used to acquire the data and the Western blot shown in Figure 1C is cropped to show the include higher molecular mass bands.

Reviewer #3 (Remarks to the Author):

Overall the authors have made very responsive changes to the manuscript which is now far clearer and important for the field.

The interpretation of the non-significant GSEA terms is the largest remaining concern. Once again, terms that are not significant to a statistically justifiable threshold cannot be used to draw conclusions. While it is promising that relevant terms arise from an unbiased analysis, none of these are altered compared to the untreated controls, therefore gene expression associated with these terms is not affected by fibril addition. I recommend that this analysis be moved into a supplemental figure and the lines from page 8 and 9:

“it is noteworthy that certain pathways were differentially up-regulated at week 1 and week 2 post seeding (Fig. 4c). Specifically, among the top 20 GO terms, Unfolded Protein Response and endoplasmic reticulum (ER) stress, endosomal and endosome to Golgi transport were up-regulated at week 1 whereas protein folding, membrane protein proteolysis, mitochondrial membrane protein insertion, organelle fusion and apoptosis were detected in week 2. These pathways that have been implicated in PD pathogenesis, suggest a temporal (i.e. time-dependent) sequence of intracellular defects around organelle membrane function that may underpin the progressive neuronal dysfunction and eventually death caused by a-synuclein aggregation.”

Need to be modified to simply state that gene expression associated with these terms is not affected by fibril addition.

Response to the Reviewers

Reviewer #1

The authors have done an admirable job at addressing my comments, and I believe that the quality of the manuscript has improved significantly.

We thank the reviewer for their comment.

I believe that it was wise to change the wording in the paper to reflect that the experiments involve "strains amplified from MSA or PD patients". However, I think the authors need to briefly mention (probably in the Discussion) the recent findings from Sjors Scheres group on the structures of brain-derived vs. amplified MSA alpha-synuclein aggregates ("Seeded assembly in vitro does not replicate the structures of α -synuclein filaments from multiple system atrophy", *FEBS Open Bio*, 2021).

The study by Lovestam et al. *FEBS Open Bio*, 2021 showed that when sarkosyl-insoluble MSA fibrils are used to seed the assembly of recombinant alpha-synuclein into fibrils, the resulting amplified fibrils exhibited some differences in their cryo-EM structure compared to the sarkosyl insoluble fibrils isolated from MSA brain. A notable difference between this study and our approach is that we performed the amplification using dilute brain homogenate rather than sarkosyl insoluble fibrils. As requested by the reviewer we have now added the following statement on Page 11, lines 22 to 28: "The full extent to which fibrils amplified with brain homogenate resemble the ones found in brain requires the resolution of their cryo-EM structures. Recently, the cryo-EM structure of fibrils amplified in the presence of sarkosyl-extracted fractions from MSA brain homogenates, using a method different from the one used in this study, were found to exhibit some degree of heterogeneity and differ from the twisted fibrils in the sarkosyl-extracted fractions²⁵. This result may be due to the removal by the sarkosyl of relevant factors in the brain homogenate that are required for faithful amplification."

Reviewer #2

It is my view that the authors have adequately addressed the points made by the three reviewers, with one exception. More detail is provided below.

We thank the reviewer for their comment.

The authors have acknowledged that it is not possible to conclude that disease-relevant/specific conformations/ polymorphs were templated/imprinted onto endogenous α -synuclein in their PMCA experiments and have thus removed such references. These references have been replaced with the terms "polymorphs/ strains amplified from PD/ MSA". To me, the term 'amplified from' indicates that more of something already in existence has been produced (in this case PD/ MSA strains). It would be better to use a phrase that does not imply production of the MSA/PD strain, for example, 'amplified/ produced in the presence of PD/ MSA.'

We have now changed "amplified from" to "amplified in the presence of PD/ MSA brain lysate" and clarified this explicitly both in the abstract and the section describing the generation of these assemblies. To keep the title succinct, we have also used the term "derived from" in the title which is defined as "base something on a modification of something else" or "develop from" (Oxford English Dictionary), thus not necessarily inferring replication of the same. This term is widely used in the field in this context.

The experiments using PK digestion of neuronal lysates are a good addition to the manuscript and support the hypothesis that templated seeded aggregation is occurring in their cell model. The experiments using S129A synuclein expressing cells are also a very welcome addition to the manuscript and support the hypothesis that the pS129 signal arises from cellular synuclein. The legend for Figure 1 now states the methods used to acquire the data and the Western blot shown in Figure 1C is cropped to include higher molecular mass bands.

We thank the reviewer for these comments.

Reviewer #3

Overall the authors have made very responsive changes to the manuscript which is now far clearer and important for the field.

We thank the reviewer for their comment.

The interpretation of the non-significant GSEA terms is the largest remaining concern. Once again, terms that are not significant to a statistically justifiable threshold cannot be used to draw conclusions. While it is promising that relevant terms arise from an unbiased analysis, none of these are altered compared to the untreated controls, therefore gene expression associated with these terms is not affected by fibril addition. I recommend that this analysis be moved into a supplemental figure and the lines from page 8 and 9: “it is noteworthy that certain pathways were differentially up-regulated at week 1 and week 2 post seeding (Fig. 4c). Specifically, among the top 20 GO terms, Unfolded Protein Response and endoplasmic reticulum (ER) stress, endosomal and endosome to Golgi transport were up-regulated at week 1 whereas protein folding, membrane protein proteolysis, mitochondrial membrane protein insertion, organelle fusion and apoptosis were detected in week 2. These pathways that have been implicated in PD pathogenesis, suggest a temporal (i.e. time-dependent) sequence of intracellular defects around organelle membrane function that may underpin the progressive neuronal dysfunction and eventually death caused by a-synuclein aggregation.” Need to be modified to simply state that gene expression associated with these terms is not affected by fibril addition.

We have now moved the relevant data to the supplement as requested (Suppl. Fig. 9) and we have replaced the paragraph with the following: “Although PD-relevant pathways in organelle homeostasis were enriched in the top 20 up-regulated GO terms at week 1 and week 2 post seeding (Suppl. Fig. 9b) gene expression associated with these terms was not significantly affected by fibril addition”